# Towards Understanding Neural Collapse: The Effects of Batch Normalization and Weight Decay

**Leyan Pan**                                                          *leyanpan@gatech.edu*
*Georgia Institute of Technology*
*Atlanta, GA, 30332*

**Xinyuan Cao**                                                          *xcao78@gatech.edu*
*Georgia Institute of Technology*
*Atlanta, GA, 30332*

**Reviewed on OpenReview:** *https://openreview.net/forum?id=eKqgCPDBFg*

## Abstract

Neural Collapse (NC) is a geometric structure observed in the terminal phase of training deep neural networks, in which features within each class collapse to a single point while class means become equally separated. We study how batch normalization (BN) and weight decay (WD) influence the emergence of NC. We prove quantitative perturbation bounds showing that, in the near-optimal loss regime, any solution of the regularized cross-entropy objective must lie close to the NC configuration. Our NC bounds depend explicitly on the WD parameter and the optimality gap. Empirically, we show that BN and appropriate WD values substantially strengthen NC during training, and that NC improves consistently as training loss decreases. These results highlight the role of feature norm control in shaping the geometric structure of deep neural network representations.

## 1 Introduction

The wide application of deep learning models has raised significant interest in theoretically understanding the mechanisms underlying their success. In particular, the generalization capability of overparameterized networks continues to escape the grasp of traditional learning theory, and the quantitative roles and impacts of widely adapted training techniques including batch normalization (**BN**, Ioffe & Szegedy (2015)) and weight decay (**WD**, Loshchilov & Hutter (2017)) remains an area of active investigation.

A promising way of mechanistically understanding neural networks is by analyzing their feature learning process. Papyan et al. (2020) observed an elegant mathematical structure in well-trained neural network classifiers, termed "Neural Collapse" (abbreviated $\mathcal{NC}$ in this work, see Figure 1 for detailed visualization.) $\mathcal{NC}$ states that after sufficient training of the neural networks: **NC1** *(Variability Collapse)*: The intra-class variability of the last-layer feature vectors tends to be zero; **NC2** *(Convergence to Simplex ETF)*: The mean of the class feature vectors become equal-norm and form a Simplex Equiangular Tight Frame (ETF) around the center up to re-scaling; **NC3** *(Self-Duality)*: The last layer weights converge to match the class mean features up to re-scaling; **NC4** *(Convergence to NCC)*: The last layer of the network behaves the same as "Nearest Class Center".

These observations reveal compelling insights into the symmetry and mathematical preferences of over-parameterized neural network classifiers. Subsequently, further work has demonstrated that $\mathcal{NC}$ may play a significant role in the generalization, transfer learning (Galanti et al. (2022b); Munn et al. (2024)), depth minimization (Galanti et al. (2022a)), and implicit bias of neural networks (Poggio & Liao (2020)).

Our paper is motivated by the following two questions:

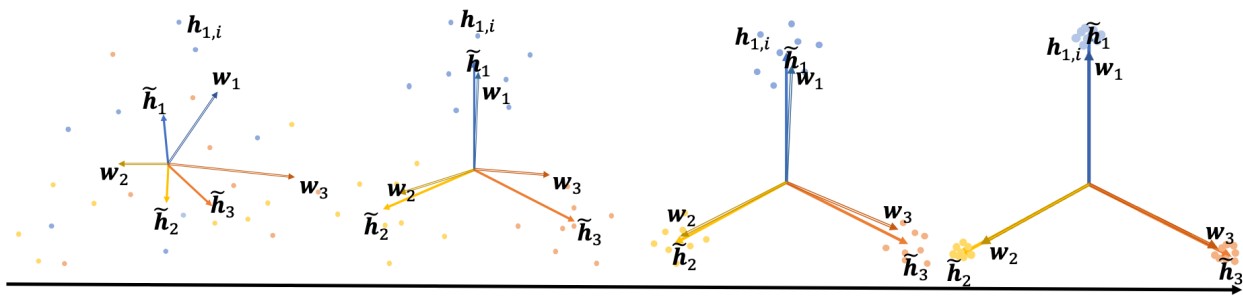

Training Time

Figure 1: Visualization of $\mathcal{NC}$ (Papyan et al. (2020)). We use an example of three classes and denote the last-layer features $\mathbf{h}_{c,i}$, mean class features $\tilde{\mathbf{h}}_c$, and last-layer class weight vectors $\mathbf{w}_{c,i}$. Circles denote individual last-layer features, while compound and filled arrows denote class weight and mean feature vectors, respectively. As training progresses, the last-layer features of each class collapse to their corresponding class means (NC1), different class means converge to the vertices of the simplex ETF (NC2), and the class weight vector of the last-layer linear classifier approaches the corresponding class means (NC3).

> - What are sufficient conditions that would guarantee proximity to the $\mathcal{NC}$ structure?
>
> - Can $\mathcal{NC}$ provide new insight into understanding how widely used training techniques, such as batch normalization and weight decay affect representation geometry?

Our work addresses these questions through an integration of theoretical analysis and empirical evidence. We first establish quantitative perturbation bounds showing that any solution with near-optimal regularized cross-entropy loss must exhibit proximity to the $\mathcal{NC}$ structure, with the tightness depending on the WD parameter and feature norm control (achieved via BN). These theoretical findings are complemented by extensive experimental investigations in both synthetic and real-world data that explore the empirical roles of BN and WD in the emergence of $\mathcal{NC}$. Consequently, our results illuminate new perspectives on understanding $\mathcal{NC}$ and the interplay between feature norm regularization and neural network training.

## 1.1 Main Results

Our theoretical results are built upon the layer-peeled model (also known as the unconstrained features model) from prior research (Mixon et al. (2022)), which considers only the last-layer features and "peels off" all other layers. In this abstraction, the hidden-layer parameters are treated as fixed, and optimization is performed only over the last-layer weights $\mathbf{W}$ and features $\mathbf{h}$, which are treated as free (unconstrained) variables. One fundamental rationale is that modern deep networks are often highly overparameterized, enabling them to learn any representation and allowing last-layer features to approximate or interpolate any point in the feature space.

We consider deep neural networks trained using cross-entropy (CE) loss on a balanced dataset. Our asymptotic theoretical analysis shows that last-layer *feature norm control (via BN with weight decay) and near-optimal cross-entropy loss* constitute sufficient conditions for proximity to several core properties of $\mathcal{NC}$. Specifically, the bounds on $\mathcal{NC}$ deviation tighten with a larger WD parameter (which constrains the feature norms) and smaller optimality gap. Complementary experiments show empirically that $\mathcal{NC}$ measures are associated with lower loss, increasing weight decay, and decreasing last-layer feature norm.

To emphasize the geometric intuition of $\mathcal{NC}$, we use cosine similarity to measure the proximity to the $\mathcal{NC}$ structure. Specifically, NC1 implies that the feature vectors in each class $c$ collapse to the same vector and achieve average feature cosine similarity of features from the same class $\text{intra}_c = 1$. NC2 implies that the class feature means achieves the maximal angle configuration, and thus the inter-class feature cosine similarity for any two classes $c, c'$ satisfies $\text{inter}_{c,c'} = -\frac{1}{C-1}$ (a property of the simplex ETF structure). Our main theorem states that, in the near-optimal regime, the intra-class and inter-class cosine similarity measures of batch-normalized models, which demonstrate the feature vectors' proximity to the $\mathcal{NC}$ structure, can be

quantitatively bounded by a function of the weight decay parameter $\lambda$ and loss value $\epsilon$ (with the class number $C$ constant when given target task).

**Theorem 1.1** (Informal version of Theorem 2.2)**.** *For the layer-peeled classification model of $C$ classes with weight decay parameter $\lambda$ and cross-entropy training loss within $\epsilon$ of the optimal loss, the following holds for most classes/pairs of classes:*

*1. (NC1) The average intra-class feature cosine similarity of class $c$:*

$$intra_c \geq 1 - O\left((C/\lambda)^{O(C)}\sqrt{\epsilon}\right),$$

*2. (NC2) The average inter-class feature cosine similarity of the class pair $c, c'$:*

$$inter_{c,c'} \leq -\frac{1}{C-1} + O\left((C/\lambda)^{O(C)}\epsilon^{1/6}\right).$$

We complement the theoretical findings with experiments on both synthetic and real datasets to investigate the factors that empirically influence $\mathcal{NC}$. Consistent with the theory, we observe that BN, increased WD, and reduced training loss are associated with stronger $\mathcal{NC}$.

Our main contributions can be summarized as follows:

- $\mathcal{NC}$ **Proximity Bound under Near-optimal Loss with Cosine Similarity Measure.** By adopting the geometrically intuitive cosine similarity measure, we prove quantitative $\mathcal{NC}$ perturbation bounds in the *near-optimal regime*, which avoids less realistic assumptions of achieving exact optimal loss. Our bounds hold for a $(1 - \delta)$ fraction of classes, providing worst-class and high-probability guarantees that the global average analysis in prior work does not readily reveal.

- **Role of Feature Norm Regularization via Weight Decay and Batch Normalization.** We show that feature norm control—achieved in practice through BN with weight decay—is a key factor in the tightness of $\mathcal{NC}$ proximity bounds. Theoretically, smaller feature and weight norms (enforced by WD on the BN scale parameters) correspond to tighter bounds on $\mathcal{NC}$ deviation from the simplex ETF structure. Empirically, our findings complement the theory by showing that $\mathcal{NC}$ is most significant with BN and high WD values.

## 1.2 Related Work

**Neural Collapse.** Our work closely relates to recent studies that analyze $\mathcal{NC}$ utilizing the layer-peeled model or unconstrained feature model (Mixon et al. (2020)). Following this model, several works have demonstrated that solutions satisfying $\mathcal{NC}$ are the only global optimizers when trained using either CE (Ji et al. (2022); Zhu et al. (2021); Lu & Steinerberger (2022)) or Mean Squared Error (MSE) loss (Han et al. (2022); Zhou et al. (2022)). These results have been extended to the class-imbalanced setting under CE loss (Dang et al. (2024); Hong & Ling (2024a); Yan et al. (2024)), and Guo et al. (2025) analyze label smoothing loss through the NC framework, deriving closed-form global minimizers and showing faster convergence to NC compared to standard CE. Our work goes beyond the global optimizer by quantitatively analyzing $\mathcal{NC}$ in the near-optimal regime, and consequently studying the factors that affect $\mathcal{NC}$.

Another line of work focuses on analyzing the training dynamics and optimization landscape using the unconstrained feature model (UFM) (Mixon et al. (2020); Zhu et al. (2021); Ji et al. (2022); Han et al. (2022); Yaras et al. (2022)). These works establish that, under both CE and MSE loss, the UFM presents a benign global optimization landscape. As a result, following gradient flow or first-order optimization methods tend to yield solutions that fulfill $\mathcal{NC}$. However, the simplification inherent in the UFM introduces a significant disparity between theory and reality. Specifically, optimizing weights in the earlier layers of a network can lead to outcomes markedly different from those achieved by direct optimization of the last-layer features. Recently, Jacot et al. (2025) addressed this gap by proving that wide neural networks (not just UFMs) trained with weight decay provably exhibit $\mathcal{NC}$, providing the first such guarantee for actual deep

networks. Súkeník et al. (2025) further prove that NC is globally optimal for deep regularized ResNets and Transformers, establishing end-to-end global optimality results for modern architectures. Several works also move beyond the UFM toward data-dependent analyses: Wu & Mondelli (2025) study NC in a three-layer network under the mean-field regime, proving that near-stationary points with small loss approximately satisfy NC1 and characterizing generalization; Hong & Ling (2024b) analyze NC in shallow ReLU networks with general data. Wang et al. (2024) show that collapse metrics improve monotonically across depth in trained ResNets, connecting NC to layer-wise feature geometry. In contrast, our findings are *optimization-agnostic* and applicable when direct optimization of the last-layer features is unfeasible.

Notably, Súkeník et al. (2024) show that multi-layer $\ell_2$ regularization induces a low-rank bias in intermediate feature representations that competes with the deep $\mathcal{NC}$ (DNC) structure, demonstrating that DNC is not always optimal for the deep unconstrained features model—an observation consistent with our empirical finding that weight decay alone (without batch normalization) does not reliably promote $\mathcal{NC}$. Additionally, Munn et al. (2024) study the geometric properties of $\mathcal{NC}$ representations and their implications for transfer learning. The NC phenomenon has also been observed beyond standard classification: Wu & Papyan (2024) find that NC emerges in large language models trained with weight decay, with NC properties correlating with generalization.

| | MSE | CE | Reg. | Norm. | Opt. | Landscape | Near-Opt. |
|---|---|---|---|---|---|---|---|
| Ji et al. (2022) | | ✓ | | | ✓* | ✓* | |
| Zhu et al. (2021) | | ✓ | ✓ | | ✓ | ✓ | |
| Lu & Steinerberger (2022) | | ✓ | | ✓ | ✓ | | |
| Poggio & Liao (2020) | ✓ | | | ✓ | ✓ | ✓ | |
| Tirer & Bruna (2022) | ✓ | | ✓ | | ✓ | | |
| Súkeník et al. (2023) | ✓ | | ✓ | | ✓ | | |
| Han et al. (2022) | ✓ | | ✓ | | ✓ | ✓ | |
| Yaras et al. (2022) | | ✓ | | ✓ | ✓ | ✓ | |
| E & Wojtowytsch (2022) | | ✓ | | ✓ | ✓ | | |
| Dang et al. (2024) | | ✓ | ✓ | | ✓ | | |
| Hong & Ling (2024a) | | ✓ | ✓ | | ✓ | | |
| Súkeník et al. (2024) | ✓ | | ✓ | | ✓ | | |
| Jacot et al. (2025) | | ✓ | ✓ | | ✓ | | |
| Súkeník et al. (2025) | ✓ | ✓ | ✓ | ✓ | ✓ | | |
| Wu & Mondelli (2025) | ✓ | | | | | ✓ | |
| This Work | | ✓ | ✓ | ✓ | ✓ | | ✓ |

Table 1: Comparison with existing theoretical works on the emergence of $\mathcal{NC}$. "Reg." denotes weight or feature norm regularization assumption, "Norm." denotes weight or feature norm constraint/normalization, "Opt." denotes optimality conditions, and "Landscape" denotes landscape or gradient flow analysis. * Shows the direction of gradient flow as it tends toward infinity without normalization/regularization.

We provide a high-level comparison of our work with prior works on $\mathcal{NC}$ in table 1.

**Weight Decay.** The concept of WD or $\ell_2$ regularization originates from early research in the stability of inverse problems (Tikhonov et al. (1943)), and has since been extensively investigated in the field of statistics (Hoerl & Kennard (1970)). In the context of neural networks, WD serves as a constraint of the network capacity (Goodfellow et al. (2016)). Several studies have demonstrated that WD enhances the model generalization by suppressing irrelevant weight vector components and diminishing static noise in the targets (Krogh & Hertz (1991); Shalev-Shwartz & Ben-David (2014)). Additionally, various studies regard WD as a mechanism that favorably affects optimization dynamics. Several works contribute to the success of WD in changing the effective learning rate (Van Laarhoven (2017); Li et al. (2020a;b)). Andriushchenko et al. (2023) demonstrates that WD improves the balance in the bias-variance optimization tradeoff, which leads to lower training loss.

**Batch Normalization.** BN was first introduced by Ioffe & Szegedy (2015) to address the issue of internal covariate shift in deep neural networks. Liao & Carneiro (2016) argues that BN mitigates the ill-conditioning

problem as the network depth increases. Luo et al. (2018) decomposes BN intro population normalization and an explicit regularization. Numerous empirical studies have demonstrated BN's positive effects on the optimization landscape through large-scale experiments (Bjorck et al. (2018); Santurkar et al. (2018); Kohler et al. (2019)). Yang et al. (2019) shows that BN regularizes the gradients and improves the optimization landscape using mean field theory. More recently, Balestriero & Baraniuk (2022) explores BN from the perspective of function approximation, arguing that BN adapts the geometry of network's spline partition to match the data.

## 2 Theoretical Results

### 2.1 Problem Setup and Notations

**Neural Network with Cross-Entropy (CE) Loss.** In this work, we consider neural network classifiers without bias terms trained using CE loss on a balanced dataset. A vanilla deep neural network classifier is composed of a feature representation function $\boldsymbol{h}^{(L)}(\boldsymbol{x})$ and a linear classifier parameterized by $\mathbf{W}^{(L)}$. Specifically, an $L$-layer vanilla deep neural network can be mathematically formulated as:

$$f(\boldsymbol{x};\boldsymbol{\theta}) = \underbrace{\boldsymbol{W}^{(L)}}_{\text{Last layer weight } \mathbf{W} = \mathbf{W}^{(L)}} \underbrace{BN\left(\sigma\left(\boldsymbol{W}^{(L-1)}\cdots\sigma\left(\boldsymbol{W}^{(1)}\boldsymbol{x}\right)\right)\right)}_{\text{last-layer feature } \boldsymbol{h}=\phi_{\boldsymbol{\theta}}(\boldsymbol{x})}.$$

Each layer is composed of an affine transformation parameterized by weight matrix $\boldsymbol{W}^{(l)}$ followed by a non-linear activation $\sigma$ such as $\text{ReLU}(x) = \max\{x, 0\}$ and BN.

The network is trained by minimizing the empirical risk over all samples $\{(\boldsymbol{x}_{c,i}, \boldsymbol{y}_c)\}, c \in [C], i \in [N]$ where each class contains $N$ samples and $\boldsymbol{y}_c$ is the one-hot encoded label vector for class $c$. We also denote $\mathbf{h}_{c,i} = \boldsymbol{h}(\boldsymbol{x}_{c,i})$ as the last-layer feature corresponding to $\boldsymbol{x}_{c,i}$. The training process minimizes the average CE loss

$$\mathcal{L} = \frac{1}{CN}\sum_{c=1}^{C}\sum_{i=1}^{N}\mathcal{L}_{\text{CE}}(f(\boldsymbol{x}_{c,i};\boldsymbol{\theta}), \boldsymbol{y}_c) = \frac{1}{CN}\sum_{c=1}^{C}\sum_{i=1}^{N}\mathcal{L}_{\text{CE}}(\boldsymbol{W}\boldsymbol{h}_{c,i}, \boldsymbol{y}_c),$$

where the cross entropy loss function for a one-hot encoding $\boldsymbol{y}_c$ is:

$$\mathcal{L}_{\text{CE}}(\boldsymbol{z}, \boldsymbol{y}_c) = -\log\left(\frac{\exp(z^{(c)})}{\sum_{c'=1}^{C}\exp(z^{(c')})}\right).$$

**Batch Normalization and Weight Decay.** For a given batch of vectors $\{\mathbf{v}_1, \mathbf{v}_2, \cdots, \mathbf{v}_b\} \subset \mathbb{R}^d$, let $v^{(k)}$ denote the $k$'th element of $\mathbf{v}$. BN developed by Ioffe & Szegedy (2015) performs the following operation along each dimension $k \in [d]$:

$$BN(\mathbf{v}_i)^{(k)} = \frac{v_i^{(k)} - \mu^{(k)}}{\sigma^{(k)}} \times \gamma^{(k)} + b^{(k)}.$$

Where $\mu^{(k)}$ and $(\sigma^{(k)})^2$ are the mean and variance along the $k$'th dimension of all vectors in the batch. The vectors $\boldsymbol{\gamma}$ and $\boldsymbol{b}$ are trainable parameters that represent the desired variance and mean after BN. In our work, we consider BN layers without bias (i.e. $\boldsymbol{b} = 0$).

WD is a technique in deep learning training that regularizes neural network weights. Specifically, the Frobenius norm of each weight matrix $\boldsymbol{W}^{(l)}$ and BN weight vector $\boldsymbol{\gamma}^{(l)}$ is added as a penalty term to the final loss. Thus, the regularized loss function with WD parameter $\lambda$ is

$$\mathcal{L}_{\text{reg}} = \mathcal{L} + \frac{\lambda}{2}\sum_{l=1}^{L}(\|\boldsymbol{\gamma}^{(l)}\|^2 + \|\mathbf{W}^{(l)}\|_F^2), \tag{1}$$

We consider the simplified layer-peeled model that only applies WD regularization to the network's final linear and BN layer. Under this setting, the regularized loss is:

$$\mathcal{L}_{\text{reg}} = \mathcal{L} + \frac{\lambda}{2}(\|\boldsymbol{\gamma}\|^2 + \|\mathbf{W}\|_F^2), \tag{2}$$

where $\mathbf{W}$ is the last layer weight matrix and $\boldsymbol{\gamma}$ is the weight of the BN layer before the final linear transformation. Note that in this formalization, the regularized BN layer acts as an explicit feature norm regularizer: BN first normalizes features to zero mean and unit variance per dimension, and the $L_2$-regularized scale parameter $\boldsymbol{\gamma}$ then directly controls the norm of the resulting features ($\sqrt{\frac{1}{N}\sum_i \|\mathbf{h}_i\|^2} = \|\boldsymbol{\gamma}\|$). Thus, applying WD to $\boldsymbol{\gamma}$ is functionally equivalent to constraining the last-layer feature norms.

## 2.2 Cosine Similarity Measure of Neural Collapse

Numerous measures of NC have been used in past literature, including within-class covariance (Papyan et al. (2020)), signal-to-noise (SNR) ratio (Han et al. (2022)), as well as class distance normalized variance (CDNV, Galanti et al. (2022b)). In this work, we focus on the cosine similarity measure (Kornblith et al. (2020)) of $\mathcal{NC}$, which emphasizes simplicity and geometric interpretability at the cost of discarding norm information. Cosine similarity is widely used as a measure between features of different samples in both practical feature learning and machine learning theory.

The average intra-class cosine similarity of class $c$ is defined as:

$$intra_c = \frac{1}{N^2}\sum_{i=1}^{N}\sum_{j=1}^{N}\cos_\angle(\mathbf{h}_{c,i} - \tilde{\mathbf{h}}_G, \mathbf{h}_{c,j} - \tilde{\mathbf{h}}_G),$$

where

$$\cos_\angle(\mathbf{x}, \mathbf{y}) = \frac{\mathbf{x}^\top \mathbf{y}}{\|\mathbf{x}\| \cdot \|\mathbf{y}\|}, \quad \tilde{\mathbf{h}}_G = \underset{c,i}{\text{Avg}}\{\mathbf{h}_{c,i}\}.$$

Similarity, the inter-class cosine similarity between two classes $c, c'$ is defined as:

$$inter_{c,c'} = \frac{1}{N^2}\sum_{i=1}^{N}\sum_{j=1}^{N}\cos_\angle(\mathbf{h}_{c,i} - \tilde{\mathbf{h}}_G, \mathbf{h}_{c',j} - \tilde{\mathbf{h}}_G)$$

In our theoretical analysis, we consider batch normalized last layer features without the bias term, and thus the global mean $\tilde{\mathbf{h}}_G$ is guaranteed to be zero and thus can be discarded.

**Relationship with $\mathcal{NC}$.** While cosine similarity does not measure vector norms, it can describe *necessary* conditions for the core observations of $\mathcal{NC}$ as follows:

(NC1) *(Variability Collapse)* All features in the same class collapse to the class mean and must achieve an intra-class cosine similarity $intra_c \to 1$.

(NC2) *(Convergence to Simplex ETF)* Class means converge to the vertices of a simplex ETF, which implies that $inter_{c,c'} \to -\frac{1}{C-1}$.

(NC3) *(Convergence to Self-Duality)* Centered class weights $\dot{\mathbf{w}}_c$ and their corresponding features $\tilde{\mathbf{h}}_c$ converge to each other up to rescaling, i.e., $\cos_\angle(\dot{\mathbf{w}}_c, \tilde{\mathbf{h}}_c) \to 1$.

As Papyan et al. (2020) has shown that NC4 is a corollary of NC1-3, we will also mainly focus on NC1-3.

## 2.3 Main Results

Before presenting our main theorem (Theorem 1.1) on BN and WD, we first present a more general preliminary theorem that provides theoretical bounds for the intra-class and inter-class cosine similarity for any classifier

with near-optimal (unregularized) CE loss. Our first theorem states that if the average last-layer feature norm and the last-layer weight matrix norm are both *bounded*, then achieving *near-optimal loss* implies that *most classes* have intra-class cosine similarity near one and *most pairs of classes* have inter-class cosine similarity near $-\frac{1}{C-1}$.

**Theorem 2.1** ($\mathcal{NC}$ proximity guarantee with bounded norms). *For any neural network classifier without bias trained on a dataset with the number of classes $C \geq 3$, samples per class $N \geq 1$, and the last layer feature dimension $d \geq C$. Under the following assumptions:*

1. *The quadratic average of the last-layer feature norms $\sqrt{\frac{1}{CN} \sum_{c=1}^{C} \sum_{i=1}^{N} \|\mathbf{h}_{c,i}\|^2} \leq \alpha$.*

2. *The Frobenius norm of the last-layer weight $\|\mathbf{W}\|_F \leq \sqrt{C}\beta$.*

3. *The average cross-entropy loss over all samples $\mathcal{L} \leq m + \epsilon$ for small $\epsilon > 0$.*

*Here $m = \log(1 + (C-1)\exp(-\frac{C}{C-1}\alpha\beta))$ is the minimum achievable loss under the norm constraints. Then for at least $1 - \delta$ fraction of all classes, with $\frac{\epsilon}{\delta} \ll 1$, there is*

$$intra_c \geq 1 - O\left(\frac{e^{O(C\alpha\beta)}}{\alpha\beta} \sqrt{\frac{\epsilon}{\delta}}\right),$$

$$\cos_\angle(\dot{\mathbf{w}}_c, \tilde{\mathbf{h}}_c) \geq 1 - O(e^{O(C\alpha\beta)}\sqrt{\frac{\epsilon}{\delta}}),$$

*and for at least $1 - \delta$ fraction of all pairs of classes $c, c'$, with $\frac{\epsilon}{\delta} \ll 1$, there is*

$$inter_{c,c'} \leq -\frac{1}{C-1} + O\left(\frac{e^{O(C\alpha\beta)}}{\alpha\beta} \left(\frac{\epsilon}{\delta}\right)^{1/6}\right).$$

The quantitative bounds of our theorem show that, conditional on near-optimal loss, smaller last-layer feature and weight norms correspond to tighter proximity to the $\mathcal{NC}$ structure.

The proof of Theorem 2.1 is inspired by the optimal-case proof from Lu & Steinerberger (2022), which shows the global optimality conditions using Jensen's inequality. Our proof extends to the near-optimal case by carefully relaxing the three strict Jensen conditions into near-optimal quantitative guarantees and analyzing the dynamics between the resulting Jensen gaps. Specifically, we show in Lemma 2.1 (based on strongly convex function result from Merentes & Nikodem (2010)) that if a set of variables achieves roughly equal value on the LHS and RHS of Jensen's inequality for a strongly convex function, then the mean of every subset cannot deviate too far from the global mean.

**Lemma 2.1** (Subset mean close to global mean). *Let $\{x_i\}_{i=1}^{N} \subset \mathcal{I}$ be a set of $N$ real numbers, let $\tilde{x} = \frac{1}{N} \sum_{i=1}^{N} x_i$ be the mean over all $x_i$ and $f$ be a function that is $m$-strongly-convex on $\mathcal{I}$. If*

$$\frac{1}{N} \sum_{i=1}^{N} f(x_i) \leq f(\tilde{x}) + \epsilon,$$

*i.e., Jensen's inequality is satisfied with gap $\epsilon$, then for any subset of samples $S \subseteq [N]$, let $\delta = \frac{|S|}{N}$, there is*

$$\tilde{x} + \sqrt{\frac{2\epsilon(1-\delta)}{m\delta}} \geq \frac{1}{|S|} \sum_{i \in S} x_i \geq \tilde{x} - \sqrt{\frac{2\epsilon(1-\delta)}{m\delta}}.$$

This lemma can be a general tool to convert optimal-case conditions derived using Jensen's inequality into worst-class and high-probability proximity bounds under near-optimal conditions. The detailed proof of Theorem 2 is provided in the supplemental materials.

We now proceed to the formal version of the main theorem that theoretically demonstrates the relationship between $\mathcal{NC}$, BN, and WD.

**Theorem 2.2** (Formal Version of Theorem 1.1)**.** *For a neural network classifier without bias trained on a dataset with the number of classes $C \geq 3$ and samples per class $N \geq 1$, we consider its layer-peeled model with batch normalization before the final layer with parameter $\boldsymbol{\gamma}$, weight decay parameter $\lambda < 1/\sqrt{C}$ and regularized CE loss*

$$\mathcal{L}_{\mathrm{reg}} = \frac{1}{CN} \sum_{c=1}^{C} \sum_{i=1}^{N} \mathcal{L}_{\mathrm{CE}} \left( \boldsymbol{W} \boldsymbol{h}_{c,i}, \boldsymbol{y}_c \right) + \frac{\lambda}{2} (\|\boldsymbol{\gamma}\|^2 + \|\mathbf{W}\|_F^2)$$

*satisfying $\mathcal{L}_{\mathrm{reg}} \leq m_{\mathrm{reg}} + \epsilon$ for small $\epsilon$, where $m_{reg}$ is the minimum achievable regularized loss. Then for at least $1 - \delta$ fraction of all classes, with $\frac{\epsilon}{\delta} \ll 1$, $\epsilon < \lambda$ and for small constant $\kappa > 0$ and $\rho = (Ce/\lambda)^{\kappa C}$, the intra-class cosine similarity for class $c$*

$$intra_c \geq 1 - \frac{C-1}{C} \sqrt{\frac{128 \rho \epsilon (1 - \delta)}{\delta}}.$$

*The cosine similarity between feature and weight for class $c$*

$$\cos_{\angle}(\dot{\mathbf{w}}_c, \tilde{\mathbf{h}}_c) \geq 1 - 2 \sqrt{\frac{2 \rho \epsilon (1 - \delta)}{\delta}}.$$

*For at least $1 - \delta$ fraction of all pairs of classes $c, c'$, with $\frac{\epsilon}{\delta} \ll 1$, the inter-class cosine similarity $inter_{c,c'}$*

$$\leq -\frac{1}{C-1} + \frac{C\rho}{C-1} \sqrt{\frac{2\epsilon}{\delta}} + 4 (\rho \sqrt{\frac{2\epsilon}{\delta}})^{1/3} + \sqrt{\rho \sqrt{\frac{2\epsilon}{\delta}}}.$$

**Proof sketch.** The proof proceeds in three steps:

1. **BN enables norm control.** BN normalizes features to zero mean and unit variance per dimension. The regularized scale parameter $\boldsymbol{\gamma}$ directly controls the post-normalization feature norms: $\sqrt{\frac{1}{N} \sum_i \|\mathbf{h}_i\|^2} = \|\boldsymbol{\gamma}\|$. WD on $\boldsymbol{\gamma}$ (with parameter $\lambda$) ensures $\|\boldsymbol{\gamma}\|^2 = O(1/\lambda)$ at near-optimal loss, since otherwise the regularization penalty would dominate.

2. **Small norms imply strong local convexity.** The CE loss $-\log(\mathrm{softmax}(\mathbf{z}))$ evaluated at logits $\mathbf{z} = \mathbf{W}\mathbf{h}$ has a local strong convexity constant that *increases* as the logit magnitudes decrease. When BN + WD keep $\|\mathbf{W}\|$ and $\|\mathbf{h}\|$ small, the loss landscape near the optimum becomes more strongly curved, constraining any near-optimal solution more tightly.

3. **Strong convexity bounds NC deviations.** Using Jensen's inequality for strongly convex functions (Lemma 2.1), a small gap $\epsilon$ from the optimal loss forces the features to concentrate: intra-class features must be nearly collinear (NC1), and class means must approach the simplex ETF configuration (NC2). The rate $\rho = (Ce/\lambda)^{O(C)}$ captures how $\lambda$ controls the tightness of these bounds through the norm constraints.

Without BN (or an equivalent mechanism that controls feature norms), the feature magnitudes remain effectively unconstrained even in the presence of WD, so the first step of the argument no longer holds and the resulting bounds become vacuous. When the BN scale parameters are learnable and regularized via WD, the combination of BN and WD on the scale parameters effectively enforces feature norm control, with the scaling parameters $\boldsymbol{\gamma}$ determining the feature magnitude. The detailed proof is provided in the supplemental materials.

Moreover, since $\rho = (Ce/\lambda)^{\kappa C}$ is a decreasing function of $\lambda$, larger values of the WD parameter lead to tighter bounds. In particular, for a fixed optimality gap $\epsilon$, the bounds imply that $intra_c$ must lie closer to 1 and $inter_{c,c'}$ closer to $-\frac{1}{C-1}$ for a $(1 - \delta)$ fraction of classes or class pairs. Intuitively, stronger WD restricts the magnitude of both the feature vectors and the classifier weights through the regularization penalty. This increases the local strong convexity of the cross-entropy loss around its optimum, leaving less room for solutions to deviate from the $\mathcal{NC}$ configuration while still achieving near-optimal loss.

Taken together, our results show that near-optimal regularized cross-entropy loss together with feature norm control provides sufficient conditions for approximate neural collapse. Importantly, the bounds are optimization-agnostic: they apply to any last-layer features and classifier weights satisfying the loss condition, regardless of the training algorithm or the structure of earlier network layers. In the next section, we complement these theoretical guarantees with empirical results examining how BN and WD influence the emergence of neural collapse in practice.

## 3 Empirical Results

In this section, we present extensive experiments that complement our theoretical analysis. Our theory establishes perturbation bounds showing that near-optimal solutions with feature norm control must lie close to the $\mathcal{NC}$ configuration. The experiments investigate whether these conditions arise in practice and how BN and WD influence the emergence of $\mathcal{NC}$ during training.

Overall, the empirical results are consistent with the theoretical predictions and reveal several qualitative trends:

- $\mathcal{NC}$ is strongest in models with BN combined with sufficiently large WD.

- Under BN, the degree of $\mathcal{NC}$ increases steadily as the training loss decreases.

- Stronger feature norm control (smaller last-layer feature norms) leads to stronger $\mathcal{NC}$.

### 3.1 Setup

We perform experiments on both synthetic and real-world datasets.

**Synthetic Datasets.** Our first set of experiments uses a vanilla neural network (i.e., Multi-Layer Perceptron with ReLU activation) to classify well-defined synthetic datasets of different distribution complexities. We aim to use straightforward model architectures and well-defined distributions to explore the effect of different hyperparameters in $\mathcal{NC}$ under a controlled setting. We consider MLP models with and without BN. In BN models, one BN layer is located after the last ReLU activation and before the final linear transformation.

Our first dataset is the conic hull dataset, where the feature space $\mathbb{R}^d$ is separated into $C$ classes using $\lceil \log C \rceil$ randomly generated hyperplanes. Since every pair of classes is linearly separable, neural networks can find a set of weights that perfectly classify all data. Thus, the conic hull dataset is a great starting point for understanding deep classification models. In our experiments, we use class number $C = 4$, dimension $d = 16$, and training dataset size $N = 8000$.

We also perform experiments on a more complex dataset where the class labels are generated using a randomly initialized MLP. We ensure that the number of layers and parameters within this data-generator MLP is less than any model used for training. The number of classes, dimensions, and training samples we use are identical to the conic hull dataset.

**Real-World Datasets. (Results in Appendix)** Our next set of experiments explores the effect of BN and WD using standard computer vision datasets MNIST (LeCun et al. (2010)), CIFAR-10, CIFAR-100 (Krizhevsky (2009)), and ImageNet32 (Deng et al. (2009)). We use VGG11 and VGG19 (Simonyan & Zisserman (2015)) convolutional neural networks as the architecture. Similar to the synthetic experiments, we consider the models with and without BN. The BN model incorporates a BN layer after selected convolution layers. Both models are official implementations of the PyTorch Library.

**Measures of proximity to the $\mathcal{NC}$ structure.** Our experiments adopt the geometrically intuitive cosine similarity measure of $\mathcal{NC}$ as in our theoretical results. While most prior empirical works of $\mathcal{NC}$ focus on the average measures of NC over all classes, (e.g., Papyan et al. (2020); Ji et al. (2022)), we additionally measure the stricter *minimum* intra-class and *maximum* inter-class (i.e. the **worst-case** measure over all classes/pairs of classes). When the number of classes is large, the difference between the average and worst-case measures

can be very significant and reveal further insights into the details of the feature geometric configuration, as later demonstrated in our experiments. Our primary empirical focus is on NC1 and NC2, as these provide the most direct geometric interpretation of feature collapse. For completeness, we also report NC3 self-duality measurements later in Section 3.5.

## 3.2 Relationship with the Presence of BN and WD

In our first set of experiments, we explore the degree of $\mathcal{NC}$ under different presences of BN and values of WD. We conduct experiments on both synthetic and real-world data as described in section 3.1 with WD values varying between $10^{-4}$ and $10^{-2}$. Our experimental results for synthetic datasets are presented in Figure 2, while those for real-world datasets can be found in the supplemental materials.

Our experiments show that, in both synthetic and realistic scenarios, the highest level of $\mathcal{NC}$ is achieved by models with BN and appropriate WD. Moreover, BN allows the degree of $\mathcal{NC}$ to increase smoothly along with the increase of WD within the range of perfect interpolation, while the degree of $\mathcal{NC}$ is unstable or decreases with the increase of WD in non-BN models. Such a phenomenon is also more pronounced in simpler neural networks and easier classification tasks than in realistic classification tasks.

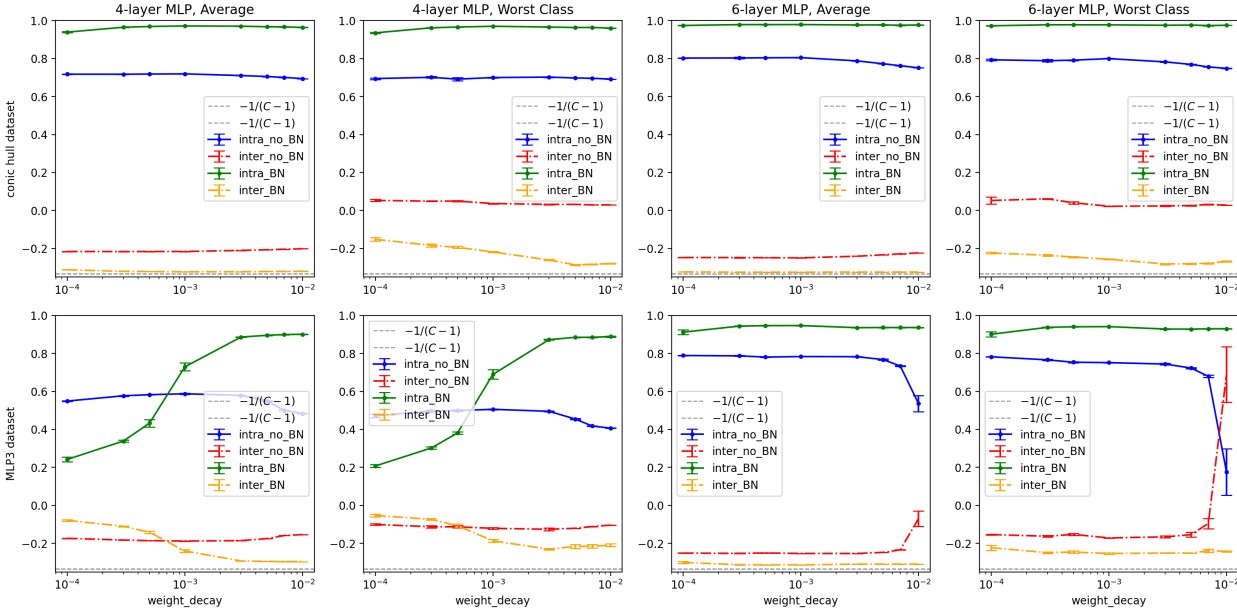

Figure 2: $\mathcal{NC}$ increases with WD under BN: Minimum intra-class and maximum inter-class Cosine Similarity for 4-layer and 6-layer MLP under Different WD and BN on the synthetic dataset generated using a randomly initialized 3-layer MLP. Higher values of intra-class and lower values of inter-class cosine similarity imply a higher degree of Neural Collapse. The **green** and **yellow** lines are cosine similarity measures for the model with BN, which demonstrates stronger $\mathcal{NC}$ along with higher WD values. The dashed gray line marks $-1/(C-1)$, the average inter-class cosine of a simplex ETF; note that reaching this average alone does not imply NC2, as it only guarantees that the class-mean unit vectors sum to zero—individual pairwise cosines can deviate from $-1/(C-1)$. Standard deviation over 5 experiments.

## 3.3 Relationship with Training Loss

Our next set of experiments explores the emergence of $\mathcal{NC}$ as the training loss decreases during the training process. Specifically, we focus on the evolution of minimum intra-class and maximum inter-class cosine similarity during training. Theorem 2.2 implies that, under feature norm control (via BN and WD), the permissible deviation from $\mathcal{NC}$ tightens as the optimality gap $\epsilon$ shrinks. However, the theory does not make predictions for the BN-free setting. We therefore investigate whether BN models exhibit a stronger empirical

correlation between loss decrease and $\mathcal{NC}$ emergence. Specifically, we record the models' cosine similarity measure every five epochs during training for both models with and without BN.

We present our results in Figure 3. We note that for the synthetic dataset experiment with BN, the degree of $\mathcal{NC}$ demonstrates a strong correlation with training loss (purple dashed line) throughout the training process while the model without BN observes little change in the $\mathcal{NC}$ beyond the first few epochs even though the loss keeps decreasing later on into the training process. For real-world experiments, the model with BN continues to demonstrate a significant correlation between training loss and $\mathcal{NC}$, while the model without BN observes an increase (instead of the expected decrease) in maximum inter-class cosine similarity during the first phases of training despite a decrease in training loss. Additional experiments with synthetic data under different WD values and real-world data are in the supplemental materials.

### 3.4 Relationship with Feature Norm

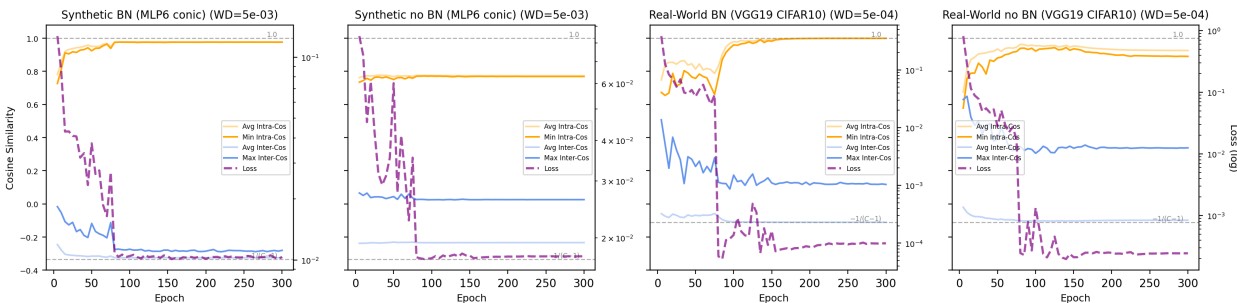

Figure 3: $\mathcal{NC}$ closely tracks loss under BN: Training curves showing cosine similarity measures (left y-axis) and training loss on a log scale (right y-axis, dashed purple line) over epochs for a representative weight decay value. Panels 1–2: synthetic data (6-layer MLP, conic hull). Panels 3–4: real-world data (VGG19, CIFAR-10). BN models (panels 1, 3) show NC metrics closely correlated with loss decrease, while no-BN models (panels 2, 4) show weaker NC improvement despite similar loss reduction. The dashed gray line marks $-1/(C-1)$.

Theorem 2.1 implies that a smaller feature norm (i.e. $\alpha$) corresponds to tighter perturbation bounds on $\mathcal{NC}$. To verify this through experiments, we fix the norm of the BN scale parameter $\boldsymbol{\gamma}$ (in (2)) to a constant value during training and compare $\mathcal{NC}$ across different $|\boldsymbol{\gamma}| := \|\boldsymbol{\gamma}\|$ values, using a WD factor of 0.005. We perform this experiment only on synthetic data (single BN layer before the final linear layer) because real-world architectures contain multiple BN layers, and fixing only the last one does not cleanly isolate the predicted effect. We vary the constant from 0.02 to 1.

As shown in Figure 4, smaller $|\boldsymbol{\gamma}|$ consistently yields stronger $\mathcal{NC}$ (middle and bottom rows). At the same time, the top row shows that smaller $|\boldsymbol{\gamma}|$ leads to a *higher* optimal loss gap $\epsilon := L_{\text{actual}} - m(\alpha\beta)$, where $m(\alpha\beta)$ is the minimum CE loss achievable at the simplex ETF given the actual feature norm $\alpha$ and per-class weight norm $\beta$ (Theorem 2.1). This indicates that the stronger $\mathcal{NC}$ observed at smaller $|\boldsymbol{\gamma}|$ cannot be explained by improved data fitting or faster optimization—in fact, convergence is slightly *worse*. Instead, it is consistent with the interpretation that tighter norm control directly promotes proximity to the $\mathcal{NC}$ geometry. Additional experiments with different WD values are provided in the supplementary material.

### 3.5 NC3 Self-Duality During Training

To complement the NC1 and NC2 measurements, we also examine the NC3 self-duality metric $\cos_{\angle}(\mathbf{w}_c, \tilde{\mathbf{h}}_c)$ during training. NC3 measures the alignment between classifier weight vectors and centered class means, where values approaching 1 indicate perfect self-duality. Although our theoretical results focus on NC1 and NC2, NC3 is known to follow from them in the near-optimal regime; empirically verifying NC3 therefore provides an additional consistency check for the collapse geometry.

Figure 5 plots the NC3 metric over training epochs. We observe that NC3 increases more rapidly and converges to values closer to 1 with BN, especially at higher WD values. Without BN, NC3 remains noticeably lower throughout training, indicating weaker duality.

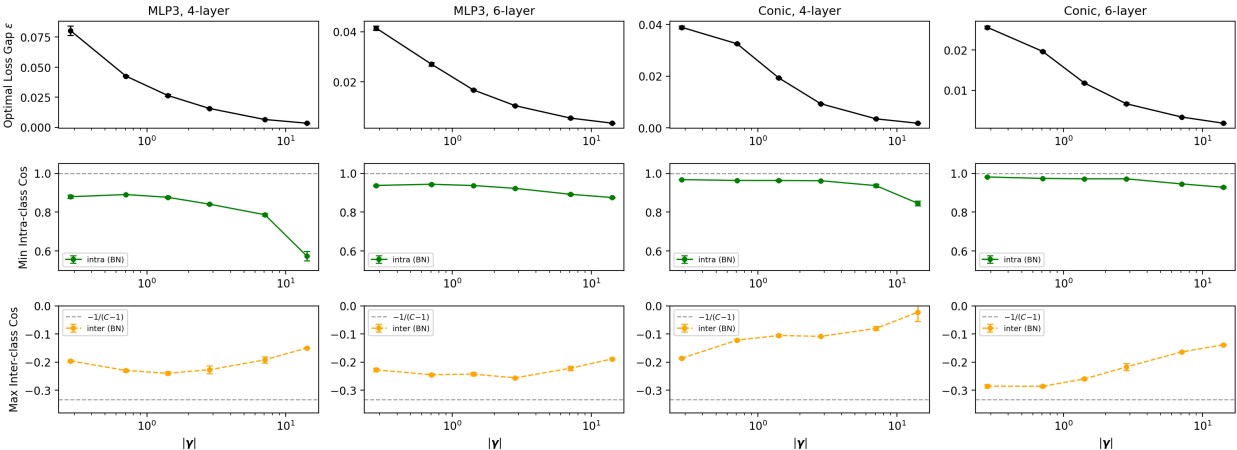

Figure 4: Fixed-$|\boldsymbol{\gamma}|$ experiments (WD$= 5\times10^{-3}$, BN, synthetic data). Top: optimal loss gap $\epsilon = L_{\text{actual}} - m(\alpha\beta)$, where $m(\alpha\beta)$ is the minimum CE loss at the simplex ETF given actual norms. Middle: min intra-class cosine similarity. Bottom: max inter-class cosine similarity. Smaller $|\boldsymbol{\gamma}|$ yields stronger $\mathcal{NC}$ (middle, bottom rows), yet the optimal loss gap *increases* (top row), ruling out better convergence as an explanation for stronger $\mathcal{NC}$. The dashed gray lines mark 1.0 (perfect NC1) and $-1/(C-1)$ (simplex ETF).

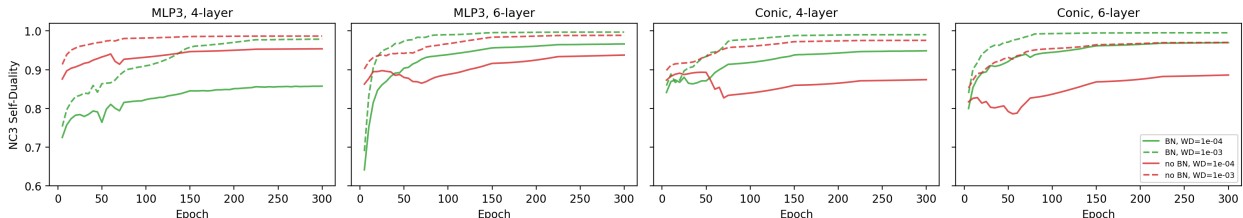

Figure 5: NC3 self-duality $\cos_{\angle}(\mathbf{w}_c, \tilde{\mathbf{h}}_c)$ during training. Values closer to 1 indicate stronger alignment between classifier weights and class means. BN models (top row) converge faster and achieve stronger self-duality, especially under larger WD values, while no-BN models (bottom row) remain farther from perfect alignment.

## 4 Limitations and Future Work

While our results provide theoretical insight into the emergence of $\mathcal{NC}$, our analysis has several limitations that suggest directions for future work.

- **Extension to deeper layers.** Our theoretical analysis follows the layer-peeled (unconstrained features) framework and focuses on the geometry of the last-layer features, with BN and WD applied at the penultimate layer. However, $\mathcal{NC}$ has also been empirically observed in deeper network layers (Ben-Shaul & Dekel, 2022; Galanti et al., 2022a), and recent theoretical work shows that $\mathcal{NC}$ can arise in deeper unconstrained feature models under regularized MSE loss (Tirer & Bruna, 2022; Súkeník et al., 2023). Extending the proximity bounds developed in this work to deeper layers and understanding how such guarantees evolve with network depth is an important direction for future research.

- **More realistic network architectures.** Our theoretical model adopts several simplifying assumptions, including the absence of bias terms and a simplified treatment of the interaction between BN and activation layers. Extending the analysis to more realistic architectures that incorporate these elements would further strengthen the connection between theoretical guarantees and practical deep neural network training.

- **Training dynamics and optimization.** Our analysis is optimization-agnostic and characterizes the geometry of near-optimal solutions rather than the training dynamics that lead to them. Understanding how optimization algorithms interact with BN, WD, and feature norm control to drive the emergence of $\mathcal{NC}$ during training remains an interesting direction for future work.

- **Tightness of theoretical bounds.** The perturbation bounds derived in this work rely on worst-case estimates and may be conservative in practice. Developing sharper bounds that more precisely capture the empirical behavior of $\mathcal{NC}$ would further improve the theoretical understanding of this phenomenon.

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

# Supplementary Material

# A    Additional Experiments

## A.1    Experiment Details

Unless otherwise specified, all models are trained on RTX4090 GPUs with learning rate $lr = 0.001$ for CIFAR10/100 and $lr = 0.0001$ for ImageNet32, which decays by a factor of 0.1 every 1/4 of the training epochs. Experiments are trained with the Adam optimizer for 300 epochs with Cross-Entropy loss. For CIFAR100 and CIFAR10 experiments, models are trained using 8000 training samples. For ImageNet32, the training sample size is 100k.

## A.2    Relationship of $\mathcal{NC}$ with BN and WD on real-world datasets

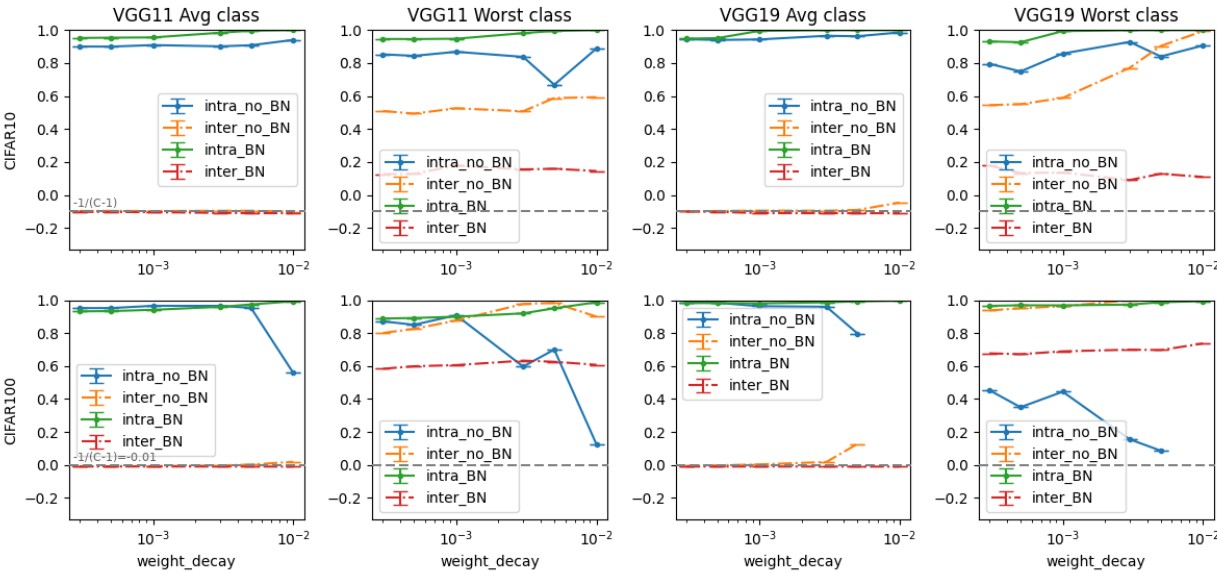

Figure 6: Intra-class and Inter-class Cosine Similarity for VGG11 and VGG19 and datasets CIFAR10 and CIFAR 100 under Different WD and BN combinations. Higher intra-class and lower inter-class cosine similarity indicate a higher degree of $\mathcal{NC}$. The dashed gray line marks $-1/(C-1)$, the simplex ETF average. Both the average measures across all classes and the worst-case measure are presented. The **green** and **red** lines are cosine similarity measures for the model with BN. Note that in the real-world data case, many models did not converge to be near-optimal, especially for the CIFAR100 dataset, on which both VGG11 and VGG19 achieve <75% accuracy. This explains why in some cases the worst-case inter-class cosine similarity can be near 1.

**Results for CIFAR10 and CIFAR 100**    In figure 6 we present experimental results for standard computer vision datasets CIFAR10 and CIFAR100 (Krizhevsky (2009)) using VGG (Simonyan & Zisserman (2015)) networks. We trained on weight decay values of $\lambda = 3e-4, 5e-4, 1e-3, 5e-3, 7e-3, 1e-2$ using two VGG implementations with and without BN in the PyTorch (Paszke et al. (2019)) library. Similar to the synthetic experiments, we consider both the average cosine similarity measures and that of the worst-performing class/pair of classes in terms of $intra_c$ and $inter_{c,c'}$ value. The **green** and **red** lines are the intra-class and inter-class cosine similarity measures for the model with BN, respectively.

We observe that, in alignment with our hypothesis, models with BN demonstrate stronger $\mathcal{NC}$ than models without BN (i.e. for intra-class, the **green** lines with BN are higher than the **blue** lines without BN, while the **red** lines for inter-class cosine similarity $inter_{c,c'}$ are above the **yellow** lines without BN). Furthermore,

$\mathcal{NC}$ is more evident as the WD value $\lambda$ increases in BN models, observable as the intra-class cosine similarity (**blue**) increases while the inter-class cosine similarity (**red**) decreases with the increase of WD value.

**Results for ImageNet32 (1000 classes).** In Figure 7, we perform experiments on the ImageNet32 dataset dataset with the VGG11, VGG19 and ResNet Model with BN. The better-performing ResNet model demonstrates the most evident $\mathcal{NC}$, which increases with the WD parameter. On the other hand, while the VGG models continue to demonstrate an increase in intra-class cosine similarity with increasing WD, the inter-class cosine similarity also increases, contrary to our theoretical prediction. This shows that optimization factors take more precedence than our optimization-agnostic theoretical bound as the number of classes $C$ increases.

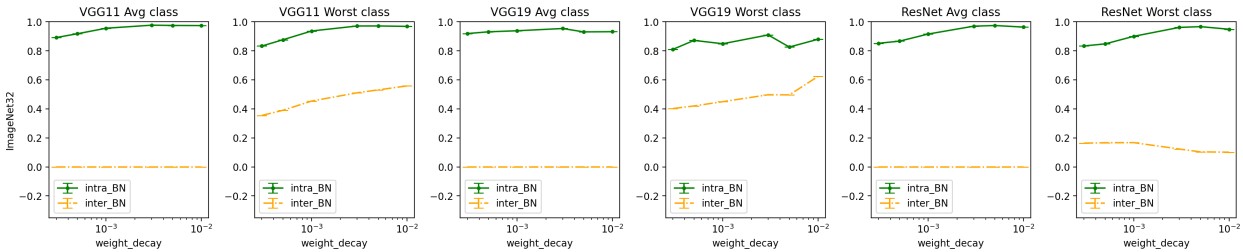

Figure 7: Intra-class and Inter-class Cosine Similarity for ImageNet32 under Different WD and BN with different models. Higher intra-class and lower inter-class cosine similarity indicate a higher degree of $\mathcal{NC}$. Both the average measures across all classes and the worst-case measure are presented. The **green** and **yellow** lines are cosine similarity measures for the model with BN.

### A.3 Relation of $\mathcal{NC}$ with training loss

In Section 3.3, we present one representative example illustrating the relationship between $\mathcal{NC}$ metrics and training loss. Figure 8 provides additional experiments on synthetic datasets and real-world datasets and models under different WD values.

Across these experiments, we consistently observe that under BN, the $\mathcal{NC}$ metrics evolve monotonically with training loss: as the loss decreases, intra-class cosine similarity increases and inter-class cosine similarity decreases toward the simplex ETF value. In contrast, without BN, the training loss may continue to decrease, whereas the $\mathcal{NC}$ metrics change little after an early stage of training.

These additional experiments therefore reinforce the observation that the relationship between decreasing loss and strengthening $\mathcal{NC}$ is substantially more reliable when BN-based feature norm control is present.

### A.4 Relation of $\mathcal{NC}$ with Last-layer Feature Norm

In Section 3.4, we examined the relationship between $\mathcal{NC}$ and the last-layer feature norm, parameterized by the norm of the batch-normalization scaling vector $\boldsymbol{\gamma}$. Those results were shown for a fixed weight decay value of $wd = 0.005$.

Figure 9 provides additional results for the same experiment across a wider range of weight decay values. Across these settings, we consistently observe that smaller $|\boldsymbol{\gamma}|$ (corresponding to smaller feature norms) leads to stronger $\mathcal{NC}$: intra-class cosine similarity increases while inter-class cosine similarity decreases toward the simplex ETF value. This trend holds across datasets, network depths, and weight decay values, indicating that the relationship between feature norm control and $\mathcal{NC}$ is robust to the choice of $wd$. At extremely small $|\boldsymbol{\gamma}|$, the model may no longer fit the training data well, which explains the slight decrease in $\mathcal{NC}$ observed at the smallest $\gamma$ values.

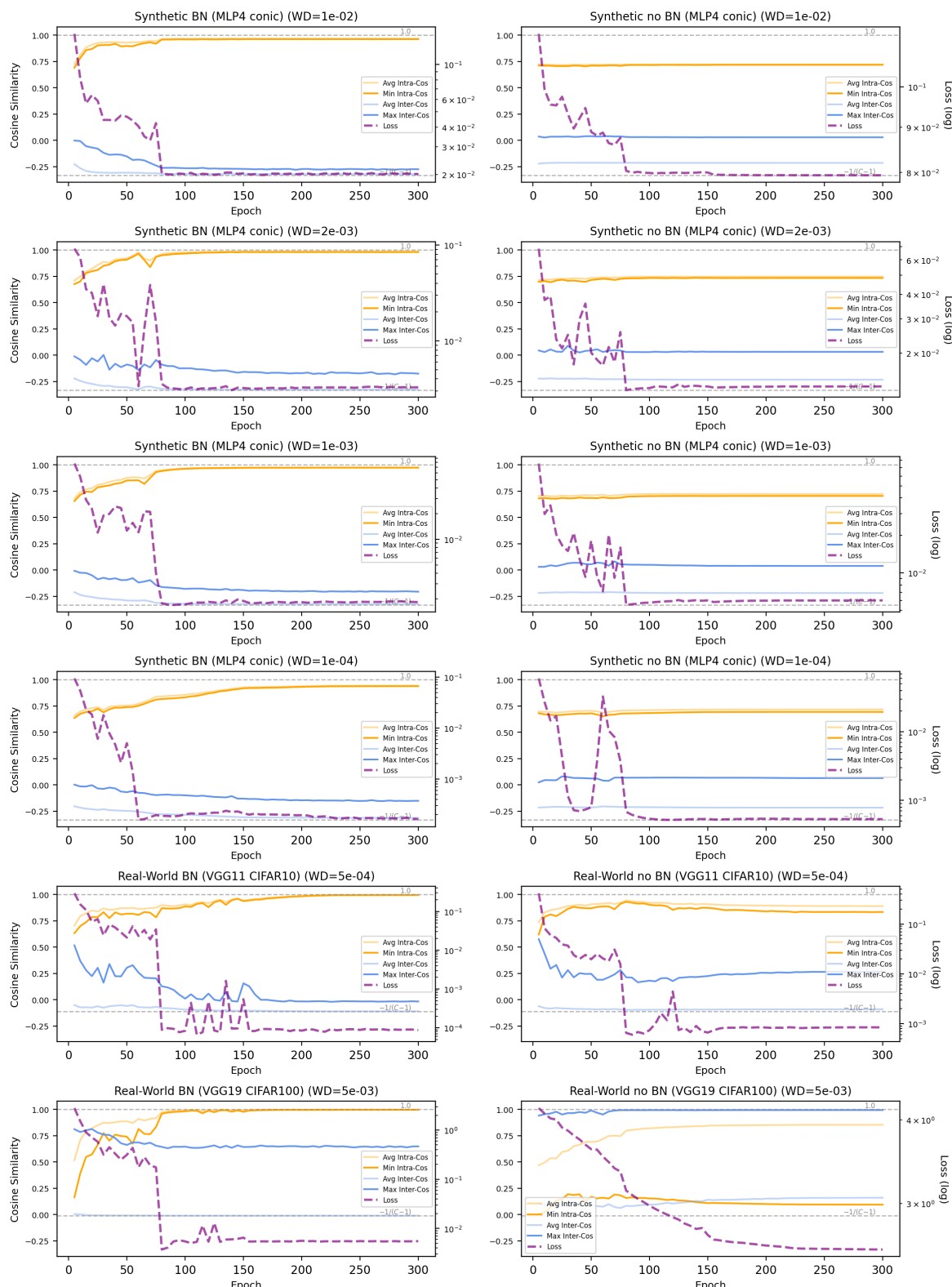

Figure 8: Minimum intra-class cosine similarity and maximum inter-class cosine similarity vs loss during training. Rows 1–4: synthetic data (4-layer MLP, conic hull) at WD values $10^{-2}, 2\times10^{-3}, 10^{-3}, 10^{-4}$. Row 5: VGG11 on CIFAR-10. Row 6: VGG19 on CIFAR-100. Left column: BN models. Right column: no-BN models. $\mathcal{NC}$ measures barely change during training without BN but increase reliably with loss decrease with BN.

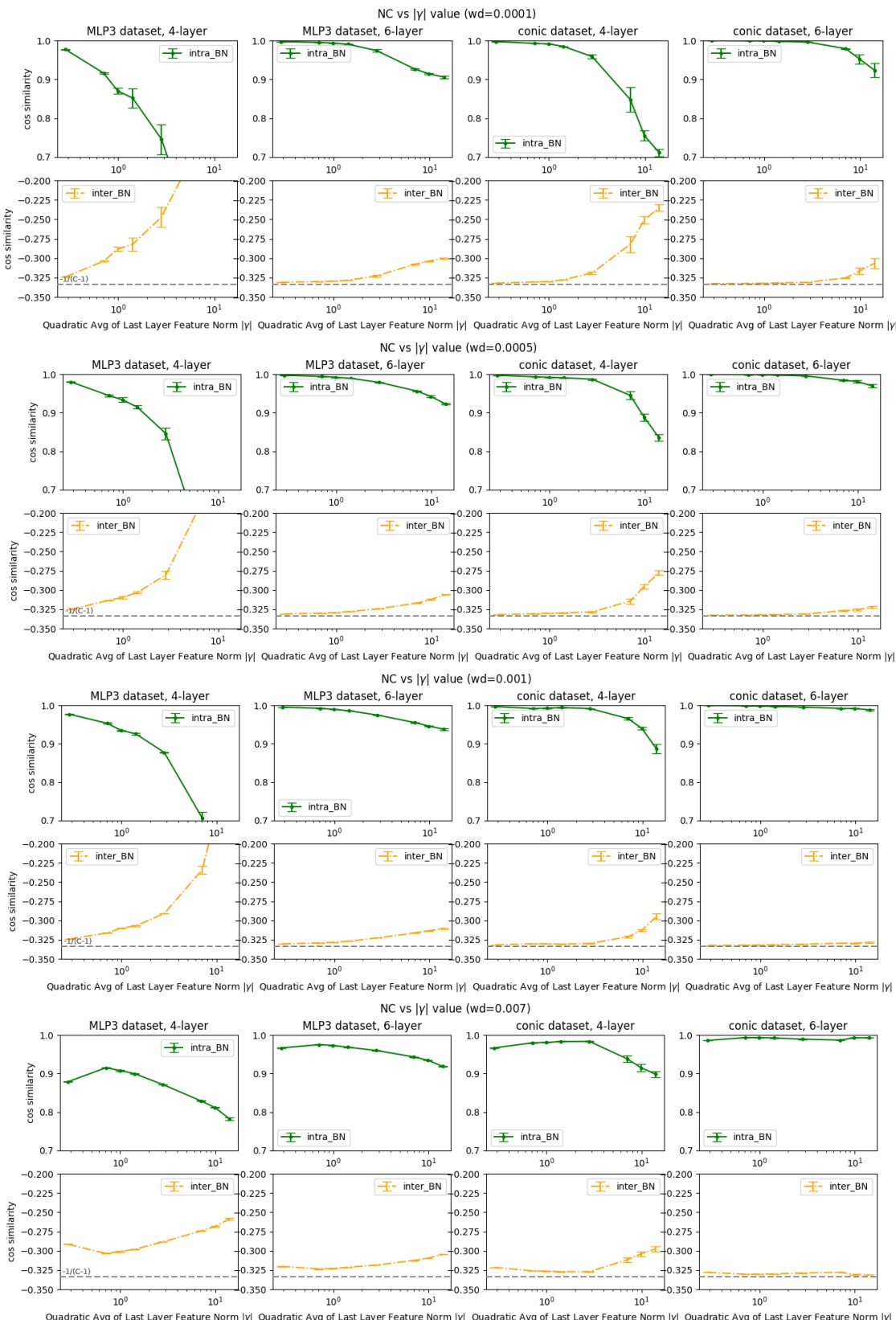

Figure 9: Relationship between $\mathcal{NC}$ and the last-layer feature norm under different WD values. Most experiments show stronger $\mathcal{NC}$ at smaller last-layer feature norms. When the feature norm becomes very small under large WD, the model may no longer closely fit the training data, which explains the slight initial decrease in $\mathcal{NC}$ observed at the smallest $\gamma$ values.

## A.5  NC Metrics vs Training Loss

Figure 10 presents a scatter plot of NC metrics sampled every 5 epochs during training against the corresponding training loss value. For BN models, both intra-class and inter-class cosine similarities show a strong, monotonic relationship with training loss—as loss decreases, NC tightens. For no-BN models, the relationship is weaker and less monotonic, especially for inter-class cosine similarity, which is consistent with our theoretical prediction that BN's feature norm control is an important mechanism underlying the perturbation bounds. Importantly, at any given loss level, BN models consistently exhibit stronger NC (higher intra-class cosine, more negative inter-class cosine) than no-BN models, confirming that BN's NC advantage is not merely a consequence of achieving lower training loss but operates through feature norm control as predicted by the theory.

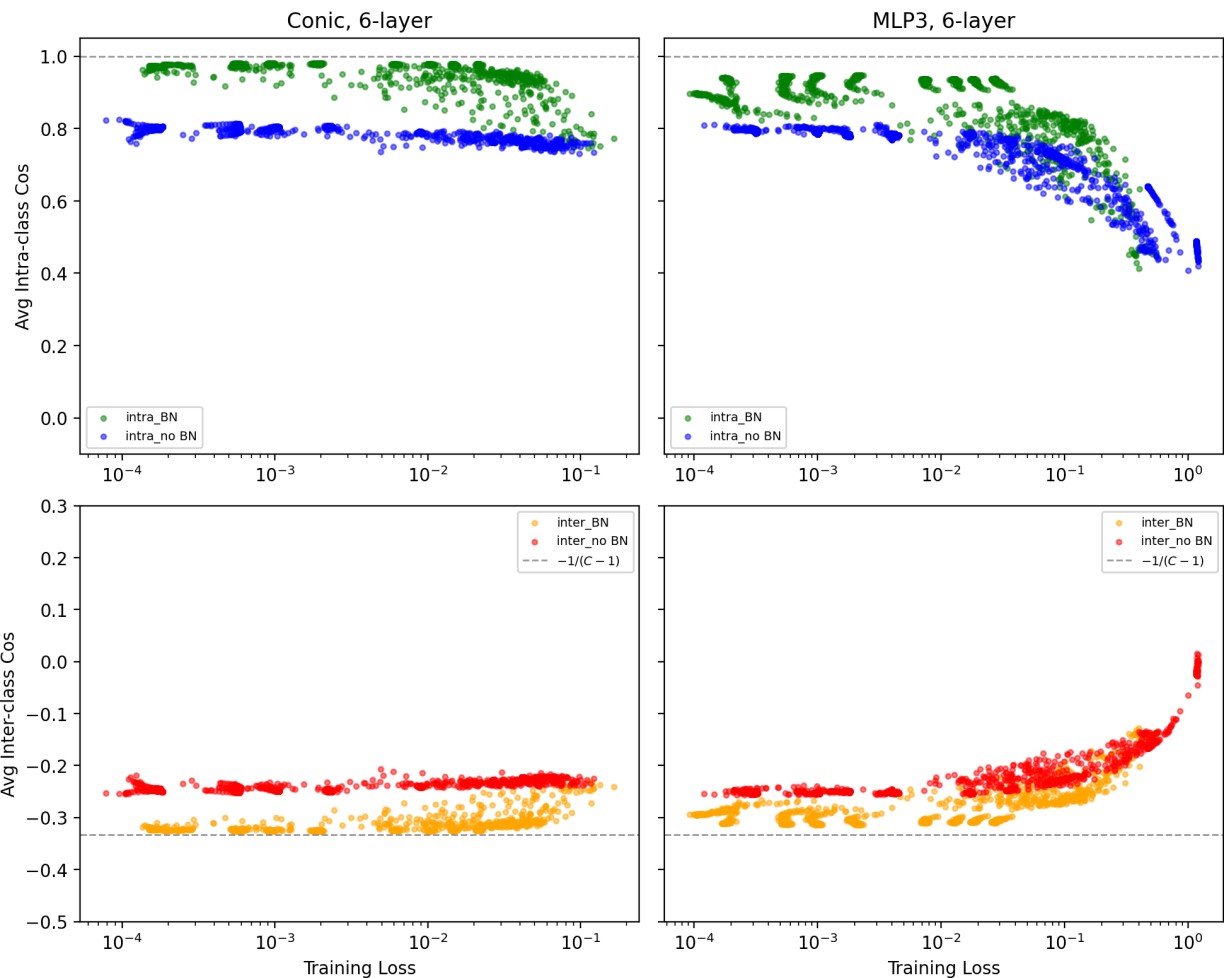

Figure 10: NC metrics vs training loss (per-epoch samples every 5 epochs). Top row: average intra-class cosine similarity. Bottom row: average inter-class cosine similarity. BN models (blue/orange) show a strong monotonic relationship between loss and NC, while no-BN models (green/red) show weaker correlation. At any given loss level, BN models consistently show stronger NC than no-BN models, confirming that BN's advantage operates beyond merely achieving lower loss. The dashed gray line marks $-1/(C-1)$.

## A.6  NC Deviation versus Optimality Gap

To more directly connect the experiments with the perturbation bounds in Theorem 2.2, we examine how the empirical deviation from $\mathcal{NC}$ varies with the training loss gap. Recall that the theorem bounds the deviation from the $\mathcal{NC}$ geometry in terms of the optimality gap $\epsilon$ and the norm-control parameter through

$\rho = (Ce/\lambda)^{\kappa C}$. This suggests that, under BN-based norm control, smaller loss gaps should correspond to smaller deviations from $\mathcal{NC}$.

Figure 11 plots the worst-class deviation metrics against the training loss $\epsilon$ on a log-log scale for 6-layer BN models. In the top row, we measure the worst intra-class deviation by $1 - \text{intra}_{\min}$; in the bottom row, we measure the worst inter-class deviation by $\text{inter}_{\max} + \frac{1}{C-1}$, i.e., the gap from the simplex ETF value. In both datasets, the deviations generally decrease as $\epsilon$ decreases, indicating that solutions with smaller loss gaps tend to lie closer to the $\mathcal{NC}$ geometry. This trend is consistent with the perturbation-based interpretation of Theorem 2.2.

Notably, we do not interpret these plots as an exact rate verification: the theoretical bounds rely on conservative worst-case estimates, so the empirical curves are better viewed as qualitative evidence that smaller optimality gaps correspond to tighter proximity to $\mathcal{NC}$.

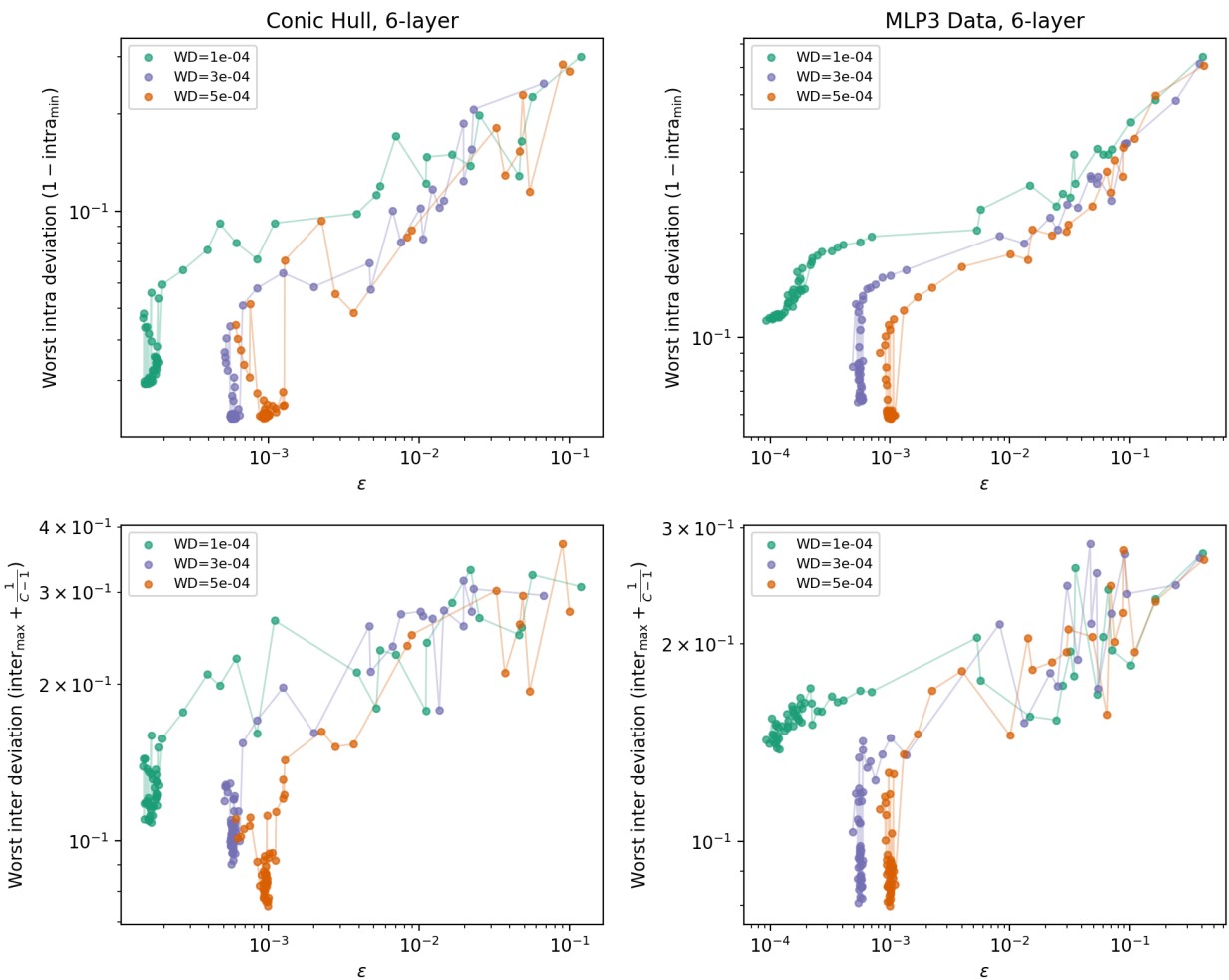

Figure 11: NC deviation versus training loss gap $\epsilon$ (log-log scale) for 6-layer BN models, shown for a single training run at each WD value. Top row: worst intra-class deviation, measured by $1 - \text{intra}_{\min}$. Bottom row: worst inter-class deviation, measured by $\text{inter}_{\max} + \frac{1}{C-1}$, i.e., the gap from the simplex ETF value. In both datasets, the deviations generally decrease as $\epsilon$ decreases, consistent with the perturbation-based prediction of Theorem 2.2 that near-optimal solutions under norm control should lie closer to the $\mathcal{NC}$ configuration.

# B    Proofs

## B.1    Proof of Lemma 2.1

Our first lemma demonstrate that if a set of variables achieves roughly equal value on the LHS and RHS of Jensen's inequality for a strongly convex function, then the mean of every subset cannot deviate too far from the global mean.

**Lemma B.1** (Restatement of lemma 2.1). *Let $\{x_i\}_{i=1}^N \subset \mathcal{I}$ be a set of $N$ real numbers, let $\tilde{x} = \frac{1}{N}\sum_{i=1}^N x_i$ be the mean over all $x_i$ and $f$ be a function that is $m$-strongly-convex on $\mathcal{I}$. If*

$$\frac{1}{N}\sum_{i=1}^N f(x_i) \leq f(\tilde{x}) + \epsilon$$

*Then for any subset of samples $S \subseteq [N]$, let $\delta = \frac{|S|}{N}$, there is*

$$\tilde{x} + \sqrt{\frac{2\epsilon(1-\delta)}{m\delta}} \geq \frac{1}{|S|}\sum_{i \in S} x_i \geq \tilde{x} - \sqrt{\frac{2\epsilon(1-\delta)}{m\delta}}$$

*Proof.* For the proof, we use a result from Merentes & Nikodem (2010) which bounds the Jensen inequality gap using the variance of the variables for strongly convex functions:

**Lemma B.2** (Theorem 4 from Merentes & Nikodem (2010)). *If $f : I \to \mathbb{R}$ is strongly convex with modulus $c$, then*

$$f\left(\sum_{i=1}^n t_i x_i\right) \leq \sum_{i=1}^n t_i f(x_i) - c\sum_{i=1}^n t_i(x_i - \bar{x})^2$$

*for all $x_1, \ldots, x_n \in I$, $t_1, \ldots, t_n > 0$ with $t_1 + \cdots + t_n = 1$ and $\bar{x} = t_1 x_1 + \cdots + t_n x_n$*

In the original definition of the authors, a strongly convex function with modulus $c$ is equivalent to a $2c$-strongly-convex function. We can apply $t_i = \frac{1}{N}$ for all $i$ and substitute the definition for strong convexity measure to obtain the following corollary:

**Corollary B.1.** *If $f : I \to \mathbb{R}$ is $m$-strongly-convex on $\mathcal{I}$, and*

$$\frac{1}{N}\sum_{i=1}^N f(x_i) = f\left(\frac{1}{N}\sum_{i=1}^N x_i\right) + \epsilon$$

*for $x_1, \ldots, x_N \in \mathcal{I}$, then $\frac{1}{N}\sum_i (x_i - \bar{x})^2 \leq \frac{2\epsilon}{m}$*

From corollary B.1, we know that $\frac{1}{N}\sum_{i=1}^n (x_i - \tilde{x})^2 \leq \frac{2\epsilon}{m}$. Let $D = \sum_{i \in S}(x_i - \tilde{x})$, by the convexity of $x^2$, there is

$$\sum_{i=1}^n (x_i - \tilde{x})^2 = \sum_{i \in S}(x_i - \tilde{x})^2 + \sum_{i \notin S}(x_i - \tilde{x})^2$$

$$\geq |S|(\frac{1}{|S|}\sum_{i \in S}(x_i - \tilde{x}))^2 + (N - |S|)(\frac{1}{N - |S|}\sum_{i \notin S}(x_i - \tilde{x}))^2$$

$$= \frac{1}{|S|}(\sum_{i \in S}(x_i - \tilde{x}))^2 + \frac{1}{N - |S|}(\sum_{i \notin S}(x_i - \tilde{x}))^2$$

$$= \frac{1}{|S|}D^2 + \frac{1}{N - |S|}(-D)^2$$

$$= \frac{D^2}{N}(\frac{1}{\delta} + \frac{1}{1-\delta})$$

$$= \frac{D^2}{N}(\frac{1}{\delta(1-\delta)})$$

Therefore $\frac{D^2}{N}(\frac{1}{\delta(1-\delta)}) \leq \frac{2\epsilon N}{m}$, and $|D| \leq \sqrt{\frac{2\epsilon\delta(1-\delta)N^2}{m}}$. Using $\frac{1}{|S|}\sum_{i\in S} x_i = \frac{1}{|S|}(|S|\tilde{x} + D)$ and $|S| = \delta N$ completes the proof. $\qquad\square$

## B.2 Proof of Theorem 2.1

We first present several lemmas that facilitate the proof technique used in the main proof. Our first lemma in this section tighens corollary B.1 specifically for the function $e^x$ and only provides the upper bound. Note that, within any predefined range $[a, b]$, $\exp(x)$ can only be guaranteed to be $e^a$ strongly convex, which may be bad if the lower bound $a$ is small or does not exist. Our further result in the following lemma shows that we can provide a better upper bound of the subset mean for the exponential function that is dependent on $\exp(\tilde{x})$ and does not require other prior knowledge of the range of $x_i$:

**Lemma B.3.** *Let $\{x_i\}_{i=1}^N \subset \mathbb{R}$ be any set of $N$ real numbers, let $\tilde{x} = \frac{1}{N}\sum_{i=1}^N x_i$ be the mean over all $x_i$. If*

$$\frac{1}{N}\sum_{i=1}^N \exp(x_i) \leq \exp(\tilde{x}) + \epsilon$$

*then for any subset $S \subseteq [N]$, let $\delta = \frac{|S|}{N}$, the there is*

$$\frac{1}{|S|}\sum_{i\in S} x_i \leq \tilde{x} + \sqrt{\frac{2\epsilon}{\delta\exp(\tilde{x})}}.$$

*Proof.* Let $D = \sum_{i\in S}(x_i - \tilde{x})$. Note that if $D < 0$ then the upper bound is obviously satisfied since the subset mean will be smaller than the global mean. Therefore, we only consider the case when $D > 0$

$$\begin{aligned}
\sum_{i=1}^N \exp(x_i) &= \sum_{i\in S}\exp(x_i) + \sum_{i\notin S}\exp(x_i) \\
&\geq |S|\exp(\frac{1}{|S|}\sum_{i\in S}x_i) + (N-|S|)\exp(\frac{1}{N-|S|}\sum_{i\notin S}x_i) \\
&\geq |S|\exp(\tilde{x} + \frac{D}{|S|}) + (N-|S|)\exp(\tilde{x} - \frac{D}{N-|S|}) \\
&\geq |S|\exp(\tilde{x})(1 + \frac{D}{|S|} + \frac{D^2}{2|S|^2}) + (N-|S|)\exp(\tilde{x})(1 - \frac{D}{N-|S|}) \\
&= (N + \frac{D^2}{2|S|})\exp(\tilde{x}) \\
N\exp(\tilde{x}) + N\epsilon &\geq (N + \frac{D^2}{2|S|})\exp(\tilde{x}) \\
D^2 &\leq \frac{2|S|N\epsilon}{\exp(\tilde{x})} \\
D &\leq N\sqrt{\frac{2\delta\epsilon}{\exp(\tilde{x})}}
\end{aligned}$$

Using $\frac{1}{|S|}\sum_{i\in S} x_i = \frac{1}{|S|}(|S|\tilde{x} + D)$ and $|S| = \delta N$ completes the proof. $\qquad\square$

Our next lemma focuses on a property of Batch Normalization: we show that BN effectively normalizes the quadratic average of the vector norms.

**Lemma B.4.** *Let $\{\mathbf{h}_i\}_{i=1}^N$ be a set of feature vectors immediately after Batch Normalization with variance vector $\boldsymbol{\gamma}$ and bias term $\boldsymbol{\beta} = 0$ (i.e. $\mathbf{h}_i = BN(\mathbf{x}_i)$ for some $\{\mathbf{x}_i\}_{i=1}^N$). Then*

$$\sqrt{\frac{1}{N}\sum_{i=1}^N \|\mathbf{h}_i\|_2^2} = \|\boldsymbol{\gamma}\|_2$$

*Proof.* Let $\boldsymbol{\gamma}$ be the variance vector for the Batch Normalization layer, and consider a single batch $\{\mathbf{x}_i\}_{i=1}^{B}$ be a batch of $B$ vectors, and

$$h_i^{(k)} = \frac{x_i^{(k)} - \tilde{x}^{(k)}}{\sigma^{(k)}} \times \gamma^{(k)}$$

for all $B$. By the linearity of mean and standard deviation, $\hat{x}_i^{(k)} = \frac{x_i^{(k)} - \tilde{x}^{(k)}}{\sigma_{\mathbf{x}}^{(k)}}$ must have mean 0 and standard deviation 1. As a result, $\sum_{i=1}^{B} \hat{x}_i^{(k)} = 0$ and $\frac{1}{B} \sum_{i=1}^{B} (\hat{x}_i^{(k)})^2 = 1$. Therefore,

$$\sum_{i=1}^{B} (h_i^{(k)})^2 = \sum_{i=1}^{B} \gamma^{(k)} (\hat{x}_i^{(k)})^2 = B(\gamma^{(k)})^2$$

$$\sum_{i=1}^{B} \|\mathbf{h}_i\|^2 = \sum_{k=1}^{d} \sum_{i=1}^{B} (h_i^{(k)})^2 = \sum_{k=1}^{d} \sum_{i=1}^{B} \gamma^{(k)} (\hat{x}_i^{(k)})^2 = \sum_{k=1}^{d} B(\gamma^{(k)})^2 = B\|\boldsymbol{\gamma}\|^2$$

Now, Consider a set of $N$ vectors divided into $m$ batches of size $\{B_j\}_{j=1}^{m}$. (This accounts for the fact that during training, the last mini-batch may have a different size than the other mini-batches if the number of training data is not a multiple of $B$). Then,

$$\sum_{i=1}^{N} \|\mathbf{h}_i\|^2 = \sum_{j=1}^{m} \sum_{i=1}^{B_j} \|\mathbf{h}_{j,i}\|^2 = \sum_{j=1}^{m} B_j \|\boldsymbol{\gamma}\|^2 = N\|\boldsymbol{\gamma}\|^2$$

Therefore, $\sqrt{\frac{1}{N} \sum_{i=1}^{N} \|\mathbf{h}_i\|^2} = \|\boldsymbol{\gamma}\|$ $\qquad\square$

Directly approaching the average intra-class and inter-class cosine similarity of vector set(s) is a relatively difficult task. Our following lemma shows that the inter-class and inter-class cosine similarities can be computed as the norm and dot product of the vectors $\tilde{\bar{\mathbf{h}}}_c$, respectively, where $\tilde{\bar{\mathbf{h}}}_c$ is the mean *normalized* vector among all vectors in a class.

**Lemma B.5.** *Let $c, c'$ be 2 classes, each containing $N$ feature vectors $\mathbf{h}_{c,i} \in \mathbb{R}^d$. Define the average intra-class cosine similarity of picking two vectors from the same class $c$ as*

$$intra_c = \frac{1}{N^2} \sum_{i=1}^{N} \sum_{j=1}^{N} \cos_\angle(\mathbf{h}_{c,i}, \mathbf{h}_{c,j})$$

*and the intra-class cosine similarity between two classes $c, c'$ is defined as the average cosine similarity of picking one feature vector of class $c$ and another from class $c'$ as*

$$inter_c = \frac{1}{N^2} \sum_{i=1}^{N} \sum_{j=1}^{N} \cos_\angle(\mathbf{h}_{c,i}, \mathbf{h}_{c',j})$$

*Let $\tilde{\bar{\mathbf{h}}}_c = \frac{1}{N} \sum_{i=1}^{N} \frac{\mathbf{h}_{c,i}}{\|\mathbf{h}_{c,i}\|}$. Then $intra_c = \|\tilde{\bar{\mathbf{h}}}_c\|^2$ and $inter_{c,c'} = \tilde{\bar{\mathbf{h}}}_c \cdot \tilde{\bar{\mathbf{h}}}_{c'}$*

*Proof.* For the intra-class cosine similarity,

$$intra_c = \frac{1}{N^2} \sum_{i=1}^{N} \sum_{j=1}^{N} \bar{\mathbf{h}}_{c,i} \cdot \bar{\mathbf{h}}_{c,j}$$

$$= \frac{1}{N^2} \sum_{i=1}^{N} \sum_{j=1}^{N} \frac{\mathbf{h}_{c,i}}{\|\mathbf{h}_{c,i}\|} \cdot \frac{\mathbf{h}_{c,j}}{\|\mathbf{h}_{c,j}\|}$$

$$= \frac{1}{N^2} \sum_{i=1}^{N} \sum_{j=1}^{N} \frac{\mathbf{h}_{c,i} \cdot \mathbf{h}_{c,j}}{\|\mathbf{h}_{c,i}\|\|\mathbf{h}_{c,j}\|}$$

$$= \left( \frac{1}{N} \sum_{i=1}^{N} \frac{\mathbf{h}_{c,i}}{\|\mathbf{h}_{c,i}\|} \right) \cdot \left( \frac{1}{N} \sum_{j=1}^{N} \frac{\mathbf{h}_{c,j}}{\|\mathbf{h}_{c,j}\|} \right)$$

$$= \|\tilde{\bar{\mathbf{h}}}_c\|^2$$

and for the inter-class cosine similarity,

$$inter_{c,c'} = \frac{1}{N^2} \sum_{i=1}^{N} \sum_{j=1}^{N} \bar{\mathbf{h}}_{c,i} \cdot \bar{\mathbf{h}}_{c',j}$$

$$= \frac{1}{N^2} \sum_{i=1}^{N} \sum_{j=1}^{N} \frac{\mathbf{h}_{c,i}}{\|\mathbf{h}_{c,i}\|} \cdot \frac{\mathbf{h}_{c',j}}{\|\mathbf{h}_{c',j}\|}$$

$$= \frac{1}{N^2} \sum_{i=1}^{N} \sum_{j=1}^{N} \frac{\mathbf{h}_{c,i} \cdot \mathbf{h}_{c',j}}{\|\mathbf{h}_{c,i}\|\|\mathbf{h}_{c',j}\|}$$

$$= \left( \frac{1}{N} \sum_{i=1}^{N} \frac{\mathbf{h}_{c,i}}{\|\mathbf{h}_{c,i}\|} \right) \cdot \left( \frac{1}{N} \sum_{j=1}^{N} \frac{\mathbf{h}_{c',j}}{\|\mathbf{h}_{c',j}\|} \right)$$

$$= \tilde{\bar{\mathbf{h}}}_c \cdot \tilde{\bar{\mathbf{h}}}_{c'}$$

$\square$

We prove the intra-class cosine similarity by first showing that the norm of the mean (un-normalized) class-feature vector for a class is near the quadratic average of feature means (i.e., $\|\tilde{\mathbf{h}}_c\| = \|\frac{1}{N} \sum_{i=1}^{N} \mathbf{h}_{c,i}\| \approx \sqrt{\frac{1}{N} \sum_{i=1}^{N} \|\mathbf{h}_{c,i}\|^2}$). However, to show intra-class cosine similarity, we need instead a bound on $\|\tilde{\bar{\mathbf{h}}}_c\| = \|\frac{1}{N} \sum_{i=1}^{N} \bar{\mathbf{h}}_{c,i}\|$ (recall that $\bar{\mathbf{v}} = \frac{\mathbf{v}}{\|\mathbf{v}\|}$ denotes the normalized vector). The following lemma provides a conversion between these requirements:

**Lemma B.6.** *Let unit vector $\mathbf{u} \in \mathbb{R}^d, \|\mathbf{u}\| = 1$, and let $\{\mathbf{v}_i\}_{i=1}^{N} \subset \mathbb{R}^d$ be a set of vectors such that $\frac{1}{N} \sum_{i=1}^{N} \|\mathbf{v}_i\|^2 \leq \alpha^2$. Define the mean of the vectors $\mathbf{v}_i$ as $\tilde{\mathbf{v}} := \frac{1}{N} \sum_{i=1}^{N} \mathbf{v}_i$.*

*Suppose that*

$$\langle \mathbf{u}, \tilde{\mathbf{v}} \rangle = \frac{1}{N} \sum_{i=1}^{N} \langle \mathbf{u}, \mathbf{v}_i \rangle \geq c,$$

*where $\frac{\alpha}{\sqrt{2}} \leq c \leq \alpha$. Define $\bar{\mathbf{v}}_i = \frac{\mathbf{v}_i}{\|\mathbf{v}_i\|}$ and $\tilde{\bar{\mathbf{v}}} := \frac{1}{N} \sum_{i=1}^{N} \bar{\mathbf{v}}_i$.*

*Then,*

$$\|\tilde{\bar{\mathbf{v}}}\| \geq 2 \left( \frac{c}{\alpha} \right)^2 - 1.$$

The proof of Lemma B.6 uses a generalization of Holder's Inequality, which we state as follows.

**Lemma B.7** (Generalized Holder's Inequality Chen (2014)). *For real positive exponents $\lambda_i$ satisfying $\lambda_a + \lambda_b + \cdots + \lambda_z = 1$, the following inequality holds.*

$$\sum_{i=1}^{n} |a_i|^{\lambda_a} |b_i|^{\lambda_b} \cdots |z_i|^{\lambda_z} \leq \left( \sum_{i=1}^{n} |a_i| \right)^{\lambda_a} \left( \sum_{i=1}^{n} |b_i| \right)^{\lambda_b} \cdots \left( \sum_{i=1}^{n} |z_i| \right)^{\lambda_z}$$

Now we are ready to prove Lemma B.6.

*Proof of Lemma B.6.* We divide all indices $i \in [N]$ into 2 sets:

$$pos = \{i \in [N] | \langle \mathbf{u}, \mathbf{v}_i \rangle \geq 0\}$$

and

$$neg = \{i \in [N] | \langle \mathbf{u}, \mathbf{v}_i \rangle < 0\}$$

Denote $I = |pos|$ as the number of indices $i$ such that $\langle u, v_i \rangle \geq 0$. We assume wlog that $pos = \{1, 2, \cdots, I\}$ and $neg = \{I+1, I+2, \cdots, N\}$. Denote $a_i = \langle u, v_i \rangle$. Then we can decompose each vector $v_i$ as follows.

$$v_i = a_i u + b_i w_i, \text{ where the unit vector } w_i \perp u, b_i \in \mathbb{R}$$

Then by normalizing $v_i$ we get

$$\bar{v}_i = \frac{a_i}{\sqrt{a_i^2 + b_i^2}} u + \frac{b_i}{\sqrt{a_i^2 + b_i^2}} w_i$$

Thus we know the vector $\tilde{\bar{v}}$ can be represented as

$$\tilde{\bar{v}} = \frac{1}{N} \sum_{i=1}^{N} \frac{a_i}{\sqrt{a_i^2 + b_i^2}} u + \frac{1}{N} \sum_{i=1}^{N} \frac{b_i}{\sqrt{a_i^2 + b_i^2}} w_i$$

Its norm can be lower bounded by

$$\|\tilde{\bar{v}}\|^2 = \left\| \frac{1}{N} \sum_{i=1}^{N} \frac{a_i}{\sqrt{a_i^2 + b_i^2}} u \right\|^2 + \left\| \frac{1}{N} \sum_{i=1}^{N} \frac{b_i}{\sqrt{a_i^2 + b_i^2}} w_i \right\|^2 \geq \frac{1}{N^2} \left( \sum_{i=1}^{N} \frac{a_i}{\sqrt{a_i^2 + b_i^2}} \right)^2$$

Take the square root of both side, and we get

$$N\|\tilde{\bar{v}}\| \geq \sum_{i=1}^{N} \frac{a_i}{\sqrt{a_i^2 + b_i^2}} = \sum_{i=1}^{I} \frac{a_i}{\sqrt{a_i^2 + b_i^2}} + \sum_{i=I+1}^{N} \frac{a_i}{\sqrt{a_i^2 + b_i^2}} \tag{3}$$

Since for any $i \geq I+1$, $i \in neg$, $a_i < 0$, and also we know for any $x, y \geq 0$,

$$\left| \frac{x}{\sqrt{x^2 + y^2}} \right| \leq 1$$

Thus for any $i \geq I+1$,

$$\frac{a_i}{\sqrt{a_i^2 + b_i^2}} \geq -1$$

By substituting this into Equation 3, we have

$$N\|\tilde{\bar{v}}\| \geq \sum_{i=1}^{N} \frac{a_i}{\sqrt{a_i^2 + b_i^2}} \geq \sum_{i=1}^{I} \frac{a_i}{\sqrt{a_i^2 + b_i^2}} - N + I \tag{4}$$

Since $\langle u, \tilde{v} \rangle \geq c$, we have

$$\sum_{i=1}^{N} a_i \geq Nc$$

Consequently,

$$\sum_{i=1}^{I} a_i > \sum_{i=1}^{N} a_i \geq Nc \tag{5}$$

We also have $\frac{1}{N}\sum_{i=1}^{N}\|v_i\|^2 \leq \alpha^2$. So we have

$$\sum_{i=1}^{I} a_i^2 \leq \sum_{i=1}^{I}(a_i^2 + b_i^2) \leq \sum_{i=1}^{N}(a_i^2 + b_i^2) \leq \sum_{i=1}^{N}\|v_i\|^2 \leq N\alpha^2 \tag{6}$$

By Cauchy-Schwarz Inequality,

$$\sum_{i=1}^{I} a_i^2 \sum_{i=1}^{I} 1 \geq \left(\sum_{i=1}^{I} a_i\right)^2$$

So we know

$$I \geq \frac{\left(\sum_{i=1}^{I} a_i\right)^2}{\sum_{i=1}^{I} a_i^2} \geq \frac{N^2 c^2}{N\alpha^2} = \frac{Nc^2}{\alpha^2} \tag{7}$$

By Lemma B.7,

$$\left(\sum_{i=1}^{I} \frac{a_i}{\sqrt{a_i^2 + b_i^2}}\right)^{2/3} \left(\sum_{i=1}^{I}(a_i^2 + b_i^2)\right)^{1/3} \geq \sum_{i=1}^{I} a_i^{2/3}$$

Combining with Equation 6, we have

$$\sum_{i=1}^{I} \frac{a_i}{\sqrt{a_i^2 + b_i^2}} \geq \sqrt{\frac{\left(\sum_{i=1}^{I} a_i^{2/3}\right)^3}{N\alpha^2}} \tag{8}$$

Then we apply Lemma B.7 again as follows.

$$\left(\sum_{i=1}^{I} a_i^{2/3}\right)^{3/4} \left(\sum_{i=1}^{I} a_i^2\right)^{1/4} \geq \sum_{i=1}^{I} a_i$$

Combining with Equation 5 and Equation 6,

$$\left(\sum_{i=1}^{I} a_i^{2/3}\right)^3 \geq \frac{(\sum_{i=1}^{I} a_i)^4}{\sum_{i=1}^{I} a_i^2} \geq \frac{N^4 c^4}{N\alpha^2} = \frac{N^3 c^4}{\alpha^2} \tag{9}$$

Using Equation 8 and Equation 9, we have

$$\sum_{i=1}^{I} \frac{a_i}{\sqrt{a_i^2 + b_i^2}} \geq \sqrt{\frac{N^2 c^4}{\alpha^4}} = \frac{Nc^2}{\alpha^2}$$

Plugging this into Equation 4 and apply Equation 7, we have

$$N\|\tilde{\tilde{v}}\| \geq \frac{Nc^2}{\alpha^2} - N + \frac{Nc^2}{\alpha^2} = N\left(\frac{2c^2}{\alpha^2} - 1\right)$$

This leads to our conclusion. $\qquad\square$

To make this lemma generalize to other proofs in future work, we provide the generalized corollary of the above lemma by setting **u** to be the normalized mean vector of **v**:

**Corollary B.2.** *Let* $\{\mathbf{v}_i\}_{i=1}^N \subset \mathbb{R}^d$ *such that* $\frac{1}{N}\|\mathbf{v}_i\|^2 \leq \alpha^2$. *If*

$$\|\tilde{\mathbf{v}}\| := \|\frac{1}{N}\sum_{i=1}^N \mathbf{v}_i\| \geq c,$$

*for* $\frac{\alpha}{\sqrt{2}} \leq c \leq \alpha$ *and let* $\bar{\mathbf{v}} := \frac{\mathbf{v}}{\|\mathbf{v}\|}$ *then*

$$\|\tilde{\bar{\mathbf{v}}}\| := \left\|\frac{1}{N}\sum_{i=1}^N \bar{\mathbf{v}}_i\right\| \geq 2\left(\frac{c}{\alpha}\right)^2 - 1.$$

*Proof.* Let $\mathbf{u} := \frac{\tilde{\mathbf{v}}}{\|\tilde{\mathbf{v}}\|}$ then $\|\mathbf{u}\| = 1$,

$$\frac{1}{N}\sum_{i=1}^N \langle \frac{\tilde{\mathbf{v}}}{\|\tilde{\mathbf{v}}\|}, \mathbf{v}_i \rangle = \langle \mathbf{u}, \tilde{\mathbf{v}} \rangle = \frac{\|\tilde{\mathbf{v}}\|^2}{\|\tilde{\mathbf{v}}\|} = \|\tilde{\mathbf{v}}\| \geq c$$

The corollary directly follows from Lemma B.6 with $\beta = \|\mathbf{u}\| = 1$ ∎

Similarly, for inter-class cosine similarity, we have the following lemma:

**Lemma B.8.** *Let* $\mathbf{w} \in \mathbb{R}^d$, $\{\mathbf{h}_i\}_{i=1}^N \subset \mathbb{R}^d$. *Let* $\tilde{\mathbf{h}} = \frac{1}{N}\sum_{i=1}^N \mathbf{h}_i$ *and* $\tilde{\bar{\mathbf{h}}} = \frac{1}{N}\sum_{i=1}^N \frac{\mathbf{h}_i}{\|\mathbf{h}_i\|}$. *If the following condition is satisfied:*

$$\mathbf{w} \cdot \tilde{\mathbf{h}} = c \qquad\qquad\qquad for\ c < 0$$
$$\|\mathbf{w}\| \leq \beta$$
$$\frac{1}{N}\sum_{i=1}^n \|\mathbf{h}_i\|^2 \leq \alpha^2$$
$$\|\tilde{\mathbf{h}}\| \geq \alpha - \frac{\epsilon}{\beta}$$
$$\epsilon \ll \alpha\beta$$

*Then* $\cos_{\angle}(\mathbf{w}, \tilde{\bar{\mathbf{h}}}) \leq -\frac{c}{\alpha\beta} + 4\left(\frac{\epsilon}{\alpha\beta}\right)^{1/3}$

*Proof.* For $\mathbf{w} \in \mathbb{R}^d$, $\{\mathbf{h}_i\}_{i=1}^N \subset \mathbb{R}^d$

Let $a_i := \frac{1}{N}\mathbf{w}\mathbf{h}_i$, $b_i := \|\mathbf{h}_i\|$, $\epsilon' := \frac{\epsilon}{\beta}$, then the constraints of the above problem can be reformulated as follows:

$$\max \sum_{i=1}^N \frac{a_i}{b_i}$$
$$s.t. \sum_{i=1}^N a_i \leq c$$
$$\frac{1}{N}\sum_{i=1}^N b_i^2 = \alpha^2$$
$$\frac{1}{N}\sum_{i=1}^N b_i \geq \alpha - \epsilon'$$
$$\forall i, |\frac{a_i}{b_i}| \leq \beta.$$

Consider a random variable $B$ that uniformly picks a value from $\{b_i\}_{i=1}^N$. Then $\mathbb{E}[B] \geq \alpha - \frac{\epsilon}{\beta}$, $\mathbb{E}[B^2] = \alpha^2$, and therefore $\sigma_B = \sqrt{\mathbb{E}[B^2] - \mathbb{E}[B]^2} \leq \sqrt{2\alpha\epsilon}$. According to Chebyshev's inequality

$$P(|B - (\alpha - \epsilon)| \geq k\sqrt{2\alpha\epsilon}) \leq \frac{1}{k^2}.$$

Note that for positive $a_i$, smaller $b_i$ means larger $\frac{a_i}{b_i}$ and for negative $a_i$, higher $b_i$ means larger $\frac{a_i}{b_i}$. Suppose that $\epsilon$ is sufficiently small such that $\epsilon \ll \sqrt{\epsilon}$. Therefore, an upper bound for $\frac{a_i}{b_i}$ when $a_i > 0$ is

$$\frac{a_i}{b_i} \leq \begin{cases} \frac{a_i}{\alpha - k\sqrt{2\alpha\epsilon}} & b_i \geq \alpha - k\sqrt{2\alpha\epsilon} \\ \beta & b_i < \alpha - k\sqrt{2\alpha\epsilon} \end{cases},$$

and an upper bound for $a_i < 0$ would is

$$\frac{a_i}{b_i} \leq \begin{cases} \frac{a_i}{\alpha + k\sqrt{2\alpha\epsilon}} & b_i \leq \alpha + k\sqrt{2\alpha\epsilon} \\ 0 & b_i > \alpha + k\sqrt{2\alpha\epsilon} \end{cases}.$$

Suppose that $k\sqrt{\frac{2\epsilon}{\alpha}}$ is less than $\frac{1}{2}$, then

$$\frac{a_i}{\alpha - k\sqrt{2\alpha\epsilon}} = \frac{a_i}{\alpha} \cdot \frac{1}{1 - k\sqrt{\frac{2\epsilon}{\alpha}}} < \frac{a_i}{\alpha} \cdot (1 + 2k\sqrt{\frac{2\epsilon}{\alpha}}) = \frac{a_i}{\alpha} + |\frac{a_i}{\alpha}| \cdot 2k\sqrt{\frac{2\epsilon}{\alpha}}$$

when $a_i > 0$, and similarly

$$\frac{a_i}{\alpha + k\sqrt{2\alpha\epsilon}} = \frac{a_i}{\alpha} \cdot \frac{1}{1 + k\sqrt{\frac{2\epsilon}{\alpha}}} < \frac{a_i}{\alpha} \cdot (1 - 2k\sqrt{\frac{2\epsilon}{\alpha}}) = \frac{a_i}{\alpha} + |\frac{a_i}{\alpha}| \cdot 2k\sqrt{\frac{2\epsilon}{\alpha}}$$

when $a_i < 0$. Note that

$$\sum_{i=1}^N |\frac{a_i}{\alpha}| \cdot 2k\sqrt{\frac{2\epsilon}{\alpha}} \leq \sum_{i=1}^N \frac{\beta}{N} \cdot 2k\sqrt{\frac{2\epsilon}{\alpha}} = 2k\beta\sqrt{\frac{2\epsilon}{\alpha}}$$

Therefore, an upper bound on the total sum would be:

$$\frac{c}{\alpha} + 2k\beta\sqrt{\frac{2\epsilon}{\alpha}} + \frac{\beta}{k^2}$$

Set $k = (\sqrt{\frac{8\epsilon}{\alpha}})^{-\frac{1}{3}}$ to get:

$$\frac{c}{\alpha} + 2\beta \left(\sqrt{\frac{8\epsilon}{\alpha}}\right)^{\frac{2}{3}} = \frac{c}{\alpha} + 4\beta \left(\frac{\epsilon}{\alpha}\right)^{\frac{1}{3}}$$

Now, we substitute $\epsilon = \frac{\epsilon'}{\beta}$ we get: $\mathbf{w} \cdot \tilde{\mathbf{h}} \leq \frac{c}{\alpha} + 4\beta \left(\frac{\epsilon'}{\alpha\beta}\right)^{1/3}$ Since $|\mathbf{w}| \leq \beta$ and $|\tilde{\mathbf{h}}| \leq 1$, we get that

$$\cos_\angle(\mathbf{w}, \tilde{\mathbf{h}}) \leq \frac{c}{\alpha\beta} + 4 \left(\frac{\epsilon'}{\alpha\beta}\right)^{1/3}$$

$\square$

**Theorem B.1** (Detailed version of Theorem 2.1). *For any neural network classifier without bias terms trained on dataset with the number of classes $C \geq 3$ and samples per class $N \geq 1$, under the following assumptions:*

*1. The quadratic average of the feature norms $\sqrt{\frac{1}{CN} \sum_{c=1}^C \sum_{i=1}^N \|\mathbf{h}_{c,i}\|^2} \leq \alpha$*

2. *The Frobenius norm of the last-layer weight $\|\mathbf{W}\|_F \leq \sqrt{C}\beta$*

3. *The average cross-entropy loss over all samples $\mathcal{L} \leq m + \epsilon$ for small $\epsilon$*

*where $m = \log(1 + (C-1)\exp(-\frac{C}{C-1}\alpha\beta))$ is the minimum achievable loss for any set of weight and feature vectors satisfying the norm constraints, then for at least $1 - \delta$ fraction of all classes , with $\frac{\epsilon}{\delta} \ll 1$, for small constant $\kappa > 0$ there is*

$$intra_c \geq 1 - \frac{C-1}{C\alpha\beta}\sqrt{\frac{128\epsilon(1-\delta)\exp(\kappa C\alpha\beta)}{\delta}} = 1 - O\left(\frac{e^{O(C\alpha\beta)}}{\alpha\beta}\sqrt{\frac{\epsilon}{\delta}}\right),$$

*and also for a cosine similarity representation of NC3 in Papyan et al. (2020):*

$$\cos_{\angle}(\dot{\mathbf{w}}_c, \tilde{\mathbf{h}}_c) \geq 1 - 2\sqrt{\frac{2\epsilon(1-\delta)e^{\kappa C\alpha\beta}}{\delta}} = 1 - O(e^{O(C\alpha\beta)}\sqrt{\frac{\epsilon}{\delta}}),$$

*and for at least $1 - \delta$ fraction of all pairs of classes $c, c'$, with $\frac{\epsilon}{\delta} \ll 1$, there is*

$$inter_{c,c'} \leq -\frac{1}{C-1} + \frac{C}{C-1}\frac{\exp(\kappa C\alpha\beta)}{\alpha\beta}\sqrt{\frac{2\epsilon}{\delta}} + 4\left(\frac{2\exp(\kappa C\alpha\beta)}{\alpha\beta}\sqrt{\frac{2\epsilon}{\delta}}\right)^{1/3} + \sqrt{\frac{\exp(\kappa C\alpha\beta)}{\alpha\beta}\sqrt{\frac{2\epsilon}{\delta}}}$$

$$= -\frac{1}{C-1} + O\left(\frac{e^{O(C\alpha\beta)}}{\alpha\beta}\left(\frac{\epsilon}{\delta}\right)^{1/6}\right)$$

*Proof.* Recall the definition of $\mathcal{L}$:

$$\mathcal{L} = \frac{1}{CN}\sum_{c=1}^{C}\sum_{i=1}^{N}\mathcal{L}_{\mathrm{CE}}\left(f(\boldsymbol{x}_{c,i}; \boldsymbol{\theta}), \boldsymbol{y}_c\right) = \frac{1}{CN}\sum_{c=1}^{C}\sum_{i=1}^{N}\mathcal{L}_{\mathrm{CE}}\left(\boldsymbol{W}\boldsymbol{h}_{c,i}, \boldsymbol{y}_c\right),$$

Let

$$L_{c,i} := \mathcal{L}_{\mathrm{CE}}\left(\boldsymbol{W}\boldsymbol{h}_{c,i}, \boldsymbol{y}_c\right)$$

denote the individual loss for sample $i$ from class $c$.

First, consider the minimum achievable average loss for a single class $c$:

$$\frac{1}{N}\sum_{i=1}^{N}L_{c,i} = -\frac{1}{N}\sum_{i=1}^{N}\log\left(\mathrm{softmax}(\mathbf{W}\mathbf{h}_{c,i})_c\right)$$

$$\geq -\log\left(\mathrm{softmax}\left(\frac{1}{N}\sum_{i=1}^{N}\mathbf{W}\mathbf{h}_{c,i}\right)_c\right)$$

$$= \log\left(1 + \sum_{c'\neq c}\exp(\frac{1}{N}\sum_{i=1}^{N}(\mathbf{w}_{c'} - \mathbf{w}_c)\mathbf{h}_{c,i})\right)$$

$$= \log\left(1 + \sum_{c'\neq c}\exp((\mathbf{w}_{c'} - \mathbf{w}_c)\tilde{\mathbf{h}}_c)\right)$$

$$\geq \log\left(1 + (C-1)\exp(\frac{1}{(C-1)}(\sum_{c'=1}^{C}\mathbf{w}_{c'}\tilde{\mathbf{h}}_c - C\mathbf{w}_c\tilde{\mathbf{h}}_c))\right)$$

$$= \log\left(1 + (C-1)\exp(\frac{1}{(C-1)}(\sum_{c'=1}^{C}\mathbf{w}_{c'} - C\mathbf{w}_c)\tilde{\mathbf{h}}_c)\right)$$

$$= \log\left(1 + (C-1)\exp(\frac{C}{C-1}(\tilde{\mathbf{w}} - \mathbf{w}_c)\tilde{\mathbf{h}}_c)\right)$$

$$= \log\left(1 + (C-1)\exp(-\frac{C}{C-1}\dot{\mathbf{w}}_c\tilde{\mathbf{h}}_c)\right)$$

Where we define $\dot{\mathbf{w}}_c := \mathbf{w}_c - \tilde{\mathbf{w}}$ Let $\overrightarrow{\mathbf{w}} = [\mathbf{w}_1 - \tilde{\mathbf{w}}, \mathbf{w}_2 - \tilde{\mathbf{w}}, \dots, \mathbf{w}_C - \tilde{\mathbf{w}}] = [\dot{\mathbf{w}}_1, \dot{\mathbf{w}}_2, \dots, \dot{\mathbf{w}}_C]$, and $\overrightarrow{\mathbf{h}} = [\tilde{\mathbf{h}}_1, \tilde{\mathbf{h}}_2, \dots, \tilde{\mathbf{h}}_c] \in \mathbf{R}^{Cd}$. Note that

$$
\begin{aligned}
\|\overrightarrow{\mathbf{w}}\|^2 &= \sum_{c=1}^{C} \|\mathbf{w}_c - \tilde{\mathbf{w}}\|^2 = \sum_{c=1}^{C} \left( \|\mathbf{w}_c\|^2 - 2\mathbf{w}_c\tilde{\mathbf{w}} + \|\tilde{\mathbf{w}}\|^2 \right) \\
&= \sum_{c=1}^{C} \|\mathbf{w}_c\|^2 - C\|\tilde{\mathbf{w}}\|^2 \leq \sum_{c=1}^{C} \|\mathbf{w}_c\|^2 = \|\mathbf{W}\|_F^2 \leq C\beta^2
\end{aligned}
$$

and also

$$
\begin{aligned}
\|\overrightarrow{\mathbf{h}}\|^2 &= \sum_{c=1}^{C} \|\tilde{\mathbf{h}}_c\|^2 = \sum_{c=1}^{C} \|\frac{1}{N}\sum_{i=1}^{N} \mathbf{h}_{c,i}\|^2 \leq \sum_{c=1}^{C} \left( \frac{1}{N}\sum_{i=1}^{N} \|\mathbf{h}_{c,i}\| \right)^2 \\
&\leq \frac{1}{N}\sum_{c=1}^{C}\sum_{i=1}^{N} \|\mathbf{h}_{c,i}\|^2 = C\alpha^2
\end{aligned}
$$

The first inequality uses the triangle inequality and the second uses $\mathbb{E}[X^2] \geq \mathbb{E}[X]^2$ Now consider the total average loss over all classes:

$$
\begin{aligned}
\mathcal{L} &= \frac{1}{CN}\sum_{c=1}^{C}\sum_{i=1}^{N} L_{c,i} \\
&\geq \frac{1}{C}\sum_{c=1}^{C} \log\left( 1 + (C-1)\exp(\frac{C}{C-1}(\tilde{\mathbf{w}} - \mathbf{w}_c)\tilde{\mathbf{h}}_c) \right) \\
&\geq \log\left( 1 + (C-1)\exp(\frac{C}{C-1} \cdot \frac{1}{C}\sum_{c=1}^{C}(\tilde{\mathbf{w}} - \mathbf{w}_c)\tilde{\mathbf{h}}_c) \right) \qquad \text{Jensen's} \\
&\geq \log\left( 1 + (C-1)\exp(-\frac{1}{C-1}\overrightarrow{\mathbf{w}} \cdot \overrightarrow{\mathbf{h}}) \right) \\
&\geq \log\left( 1 + (C-1)\exp(-\frac{C}{C-1}\alpha\beta) \right) \\
&= m,
\end{aligned}
$$

showing that $m$ is indeed the minimum achievable average loss among all samples.
Now we instead consider when the final average loss is near-optimal of value $m + \epsilon$ with $\epsilon \ll 1$. We use a new $\epsilon$ to represent the gap introduced by each inequality in the above proof. Additionally, since the average loss

is near-optimal, there must be $\dot{\mathbf{w}}_c \tilde{\mathbf{h}}_c \geq 0$ for any sufficiently small $\epsilon$:

$$\frac{1}{N}\sum_{i=1}^{N} L_{c,i} = -\frac{1}{N}\sum_{i=1}^{N} \log\left(\text{softmax}(\mathbf{W}\mathbf{h}_{c,i})_c\right) \tag{10}$$

$$\geq -\log\left(\text{softmax}\left(\frac{1}{N}\sum_{i=1}^{N}\mathbf{W}\mathbf{h}_{c,i}\right)_c\right) \tag{11}$$

$$= \log\left(1 + \sum_{c'\neq c}\exp\left(\frac{1}{N}\sum_{i=1}^{N}\mathbf{w}_{c'}\mathbf{h}_{c,i} - \frac{1}{N}\sum_{i=1}^{N}\mathbf{w}_c\mathbf{h}_{c,i}\right)\right) \tag{12}$$

$$= \log\left(1 + \sum_{c'\neq c}\exp(\frac{1}{N}\sum_{i=1}^{N}(\mathbf{w}_{c'} - \mathbf{w}_c)\mathbf{h}_{c,i})\right) \tag{13}$$

$$= \log\left(1 + \sum_{c'\neq c}\exp((\mathbf{w}_{c'} - \mathbf{w}_c)\tilde{\mathbf{h}}_c)\right) \tag{14}$$

$$= \log\left(1 + (C-1)\exp(\frac{1}{(C-1)}(\sum_{c'=1}^{C}\mathbf{w}_{c'}\tilde{\mathbf{h}}_c - C\mathbf{w}_c\tilde{\mathbf{h}}_c)) + \epsilon'_{1,c}\right) \tag{15}$$

$$= \log\left(1 + (C-1)\exp(\frac{1}{(C-1)}(\sum_{c'=1}^{C}(\mathbf{w}_{c'} - \mathbf{w}_c)\tilde{\mathbf{h}}_c) + \epsilon'_{1,c}\right) \tag{16}$$

$$= \log\left(1 + (C-1)\exp(\frac{C}{C-1}(\tilde{\mathbf{w}} - \mathbf{w}_c)\tilde{\mathbf{h}}_c) + \epsilon'_{1,c}\right) \tag{17}$$

$$\geq \log\left(1 + (C-1)\exp(-\frac{C}{C-1}\dot{\mathbf{w}}_c\tilde{\mathbf{h}}_c)\right) + \frac{\epsilon'_{1,c}}{1 + (C-1)\exp(-\frac{C}{C-1}\dot{\mathbf{w}}_c\tilde{\mathbf{h}}_c)} \tag{18}$$

$$\geq \log\left(1 + (C-1)\exp(-\frac{C}{C-1}\dot{\mathbf{w}}_c\tilde{\mathbf{h}}_c)\right) + \frac{\epsilon'_{1,c}}{C} \tag{19}$$

where $\epsilon'_{1,c} := \exp(\frac{1}{(C-1)}(\sum_{c'=1}^{C}\mathbf{w}_{c'}\tilde{\mathbf{h}}_c - C\mathbf{w}_c\tilde{\mathbf{h}}_c)) - \sum_{c'\neq c}\exp((\mathbf{w}_{c'} - \mathbf{w}_c)\tilde{\mathbf{h}}_c)$ and also

$$\mathcal{L} = \frac{1}{CN}\sum_{c=1}^{C}\sum_{i=1}^{N} L_{c,i} \tag{20}$$

$$\geq \frac{1}{C}\sum_{c=1}^{C}\left(\log\left(1 + (C-1)\exp(-\frac{C}{C-1}\dot{\mathbf{w}}_c\tilde{\mathbf{h}}_c)\right) + \frac{\epsilon'_{1,c}}{C}\right) \tag{21}$$

$$= \log\left(1 + (C-1)\exp(-\frac{C}{C-1}\cdot\frac{1}{C}\sum_{c=1}^{C}\dot{\mathbf{w}}_c\tilde{\mathbf{h}}_c)\right) + \frac{1}{C}\sum_{c=1}^{C}\frac{\epsilon'_{1,c}}{C} + \epsilon'_2 \qquad \text{Jensen's with gap } \epsilon'_2 \tag{22}$$

$$= \log\left(1 + (C-1)\exp(-\frac{1}{C-1}\overrightarrow{\mathbf{w}}\cdot\overrightarrow{\mathbf{h}})\right) + \frac{1}{C}\sum_{c=1}^{C}\frac{\epsilon'_{1,c}}{C} + \epsilon'_2 \tag{23}$$

$$= \log\left(1 + (C-1)\exp(-\frac{C}{C-1}\alpha\beta + \epsilon'_3)\right) + \frac{1}{C}\sum_{c=1}^{C}\frac{\epsilon'_{1,c}}{C} + \epsilon'_2 \tag{24}$$

where

$$\epsilon'_2 := \frac{1}{C}\sum_{c=1}^{C}\log\left(1 + (C-1)\exp(-\frac{C}{C-1}\dot{\mathbf{w}}_c\tilde{\mathbf{h}}_c)\right) - \log\left(1 + (C-1)\exp(-\frac{C}{C-1}\cdot\frac{1}{C}\sum_{c=1}^{C}\dot{\mathbf{w}}_c\tilde{\mathbf{h}}_c)\right)$$

and $\epsilon'_3 := \frac{1}{C-1}(C\alpha\beta - \overrightarrow{\mathbf{w}}\cdot\overrightarrow{\mathbf{h}})$

Consider $\log(1 + (C-1)\exp(-\frac{C\alpha\beta}{C-1} + \epsilon_3'))$: Let $\gamma' = (C-1)\exp(-\frac{C\alpha\beta}{C-1})$

$$
\begin{aligned}
\log(1 + (C-1)\exp(-\frac{C\alpha\beta}{C-1} + \epsilon_3')) &= \log(1 + (C-1)\exp(-\frac{C\alpha\beta}{C-1})\exp(\epsilon_3')) \\
&= \log(1 + (C-1)\exp(-\frac{C\alpha\beta}{C-1})\exp(\epsilon_3')) \\
&= \log(1 + \gamma'\exp(\epsilon_3')) \\
&\geq \log(1 + \gamma'(1 + \epsilon_3')) \\
&= \log(1 + \gamma' + \gamma'\epsilon_3') \\
&\geq \log(1 + \gamma') + \frac{\gamma'\epsilon_3'}{1 + \gamma' + \gamma'\epsilon_3'}
\end{aligned}
$$

Since $m + \epsilon = \log(1 + \gamma') + \epsilon \geq \log(1 + (C-1)\exp(-\frac{C\alpha\beta}{C-1} + \epsilon_3'))$, we get that $\epsilon \geq \frac{\gamma'\epsilon_3'}{1 + \gamma' + \gamma'\epsilon_3'}$, and

$$
\epsilon_3' \leq \frac{\epsilon(1 + \gamma')}{\gamma'(1 - \epsilon)} = \frac{\epsilon}{1 - \epsilon} \cdot \frac{1 + \gamma'}{\gamma'}
$$

for $\epsilon < 1$. By definition of $\epsilon_3'$, we know that

$$
\vec{\mathbf{w}} \cdot \vec{\mathbf{h}} = \sum_{c=1}^{C} \dot{\mathbf{w}}_c \tilde{\mathbf{h}}_c \geq C\alpha\beta - (C-1) \cdot \frac{\epsilon}{1 - \epsilon} \cdot \frac{1 + \gamma'}{\gamma'} = C\alpha\beta - \frac{\epsilon}{1 - \epsilon} \cdot [\exp(\frac{C\alpha\beta}{C-1}) + C - 1]
$$

For simplicity, let $\delta_2 = \frac{\epsilon}{1 - \epsilon} \cdot [\exp(\frac{C\alpha\beta}{C-1}) + C - 1]$

Since $\|\vec{\mathbf{w}}\| \leq \sqrt{C}\beta$, we know that $\|\vec{\mathbf{h}}\| \geq \sqrt{C}\alpha - \frac{\delta_2}{\sqrt{C}\beta}$ and

$$
\|\vec{\mathbf{h}}\|^2 = \sum_{c=1}^{C} \|\tilde{\mathbf{h}}_c\|^2 \geq C\alpha^2 - 2\frac{\delta_2\alpha}{\beta}.
$$

By Corollary B.2 we know that:

$$
\|\tilde{\tilde{\mathbf{h}}}_c\| \geq 2\left(\frac{\|\tilde{\mathbf{h}}_c\|}{\sqrt{\frac{1}{N}\sum_{i=1}^{N}\|\mathbf{h}_{c,i}\|}}\right)^2 - 1.
$$

Let $\alpha_c := \sqrt{\frac{1}{N}\sum_{i=1}^{N}\|\mathbf{h}_{c,i}\|^2}$, then $\sum_{c=1}^{C}\alpha_c^2 \leq C\alpha^2$.

Now, using the bound on $\|\tilde{\tilde{\mathbf{h}}}_c\|$ and the definition of $\alpha_c$, we can write:

$$
\|\tilde{\tilde{\mathbf{h}}}_c\| \geq 2\left(\frac{\|\tilde{\mathbf{h}}_c\|}{\alpha_c}\right)^2 - 1.
$$

Summing over all classes $c = 1, \ldots, C$, we get:

$$
\sum_{c=1}^{C}\|\tilde{\tilde{\mathbf{h}}}_c\| \geq \sum_{c=1}^{C}\left(2\left(\frac{\|\tilde{\mathbf{h}}_c\|}{\alpha_c}\right)^2 - 1\right).
$$

Since $\alpha_c = \sqrt{\frac{1}{N}\sum_{i=1}^{N}\|\mathbf{h}_{c,i}\|^2}$, we know that:

$$
\sum_{c=1}^{C}\alpha_c^2 \leq C\alpha^2.
$$

Hence,

$$\sum_{c=1}^{C} \|\tilde{\tilde{\mathbf{h}}}_c\| \geq 2 \sum_{c=1}^{C} \left( \frac{\|\tilde{\mathbf{h}}_c\|^2}{\alpha_c^2} \right) - C \geq 2C \left( \frac{\sum_{c=1}^{C} \|\tilde{\mathbf{h}}_c\|^2}{\sum_{c=1}^{C} \alpha_c^2} \right) - C.$$

Using the bound $\sum_{c=1}^{C} \|\tilde{\mathbf{h}}_c\|^2 \geq C\alpha^2 - 2\frac{\delta_2 \alpha}{\beta}$, we obtain:

$$\sum_{c=1}^{C} \|\tilde{\tilde{\mathbf{h}}}_c\| \geq 2C \left( \frac{C\alpha^2 - 2\frac{\delta_2 \alpha}{\beta}}{\sum_{c=1}^{C} \alpha_c^2} \right) - C.$$

Since $\sum_{c=1}^{C} \alpha_c^2 \leq C\alpha^2$, we can write:

$$\sum_{c=1}^{C} \|\tilde{\tilde{\mathbf{h}}}_c\| \geq 2C \left( \frac{C\alpha^2 - 2\frac{\delta_2 \alpha}{\beta}}{C\alpha^2} \right) - C.$$

Simplifying the expression:

$$\sum_{c=1}^{C} \|\tilde{\tilde{\mathbf{h}}}_c\| \geq 2C \left( 1 - \frac{2\frac{\delta_2 \alpha}{\beta}}{C\alpha^2} \right) - C.$$

Further simplifying:

$$\sum_{c=1}^{C} \|\tilde{\tilde{\mathbf{h}}}_c\| \geq 1 - \frac{4\delta_2}{\alpha\beta} = C - \frac{4\epsilon}{1-\epsilon} \cdot \frac{\exp(\frac{C\alpha\beta}{C-1}) + C - 1}{\alpha\beta}$$

Since each $\|\tilde{\tilde{\mathbf{h}}}_c\| \leq 1, \forall c$, we can use Markov's inequality to get that there are at least $1 - \delta$ fraction of classes for which:

$$intra_c = \|\tilde{\tilde{\mathbf{h}}}_c\| \geq 1 - \frac{4\epsilon}{(1-\epsilon)\delta} \cdot \frac{\exp(\frac{C\alpha\beta}{C-1}) + C - 1}{C\alpha\beta} = 1 - O\left( \frac{\epsilon}{\delta} \exp\left( \alpha\beta(1 + o(C)) \right) \right)$$

Thus using the fact that $1 + (C-1)\exp(-\frac{C\alpha\beta}{C-1}) \leq C$

$$\mathcal{L} \geq \log(1 + (C-1)\exp(-\frac{C\alpha\beta}{C-1})) + \frac{1}{C} \sum_{c=1}^{C} \frac{\epsilon'_{1,c}}{C} + \epsilon'_2 + \frac{\gamma'}{1+\gamma'}\epsilon'_3$$

$$\epsilon \geq \frac{1}{C} \sum_{c=1}^{C} \frac{\epsilon'_{1,c}}{C} + \epsilon'_2 + \frac{\gamma'}{1+\gamma'}\epsilon'_3$$

Note that while we do not know how $\epsilon$ is distributed among the different gaps, all the bounds involving $\epsilon'_{1,c}, \epsilon'_2, \epsilon'_3$ always hold in the worst case scenario subject to the constraint $\epsilon \geq \frac{1}{C} \sum_{c=1}^{C} \frac{\epsilon'_{1,c}}{C} + \epsilon'_2 + \frac{\gamma'}{1+\gamma'}\epsilon'_3$. Note that $\|\tilde{\mathbf{h}}_c\| \leq \sum_{c'=1}^{C} \|\tilde{\mathbf{h}}_{c'}\| \leq \sqrt{C}\alpha$, and $\|\dot{\mathbf{w}}_c\| \leq \|\mathbf{W}\|_F = \sqrt{C}\beta$ therefore $\dot{\mathbf{w}}_c\tilde{\mathbf{h}}_c \geq -C\alpha\beta$. We also know that

$$\frac{1}{C} \sum_{c=1}^{C} \dot{\mathbf{w}}_c\tilde{\mathbf{h}}_c = \frac{1}{C}\overrightarrow{\mathbf{w}} \cdot \overrightarrow{\mathbf{h}} = \frac{1}{C}(C\alpha\beta - (C-1)\epsilon'_3) = \alpha\beta - \frac{C-1}{C}\epsilon'_3$$

We now focus on the implication of $\epsilon'_2$ from (22). Note that the relaxation can be written as

$$\frac{1}{C} \sum_{c=1}^{C} \log\left( 1 + (C-1)\exp(x_c) \right) = \log\left( 1 + (C-1)\exp\left( \frac{1}{C} \sum_{c=1}^{C} x_c \right) \right) + \epsilon'_2$$

with $x_c = -\frac{C}{C-1}(\dot{\mathbf{w}}_c\tilde{\mathbf{h}}_c)$ and $\epsilon'_2 \geq 0$ because of the strong convexity of $\log(1 + (C-1)\exp(x))$. Therefore, in order to apply Lemma 2.1, we would first need to determine the degree of strong convexity of

$\log(1 + (C-1)\exp(x))$. Note that a function is $\lambda$ strongly convex if its second-order derivative is always at least $\lambda$.

The second-order derivative of $\log(1 + (C-1)\exp(x))$ is

$$\frac{(C-1)\exp(x)}{(1+(C-1)\exp(x))^2} = 1/((C-1)\exp(x) + 2 + \frac{1}{(C-1)\exp(x)}),$$

which is $e^{-\kappa C\alpha\beta}$ for any $x \in [-\frac{C^2}{C-1}\alpha\beta, \frac{C^2}{C-1}\alpha\beta]$ for small constant $\kappa$, we denote as $O(C\alpha\beta)$ further. Therefore, the function $\log(1 + (C-1)\exp(x))$ is $\lambda$-strongly-convex for $\lambda = e^{-O(C\alpha\beta)}$ Thus, for any subset $S \subseteq [C]$, let $\delta = \frac{|S|}{C}$, by Lemma 2.1:

$$-\frac{C}{C-1}\sum_{c\in S}\dot{\mathbf{w}}_c\tilde{\mathbf{h}}_c \leq \delta C(-\frac{1}{C-1}\overrightarrow{\mathbf{w}}\cdot\overrightarrow{\mathbf{h}}) + C\sqrt{\frac{2\epsilon_2'\delta(1-\delta)}{\lambda}}$$

$$\sum_{c\in S}\dot{\mathbf{w}}_c\tilde{\mathbf{h}}_c \geq \delta\overrightarrow{\mathbf{w}}\cdot\overrightarrow{\mathbf{h}} - (C-1)\sqrt{\frac{2\epsilon_2'\delta(1-\delta)}{\lambda}}$$

$$\sum_{c\in S}\alpha_c\beta_c = \sum_{c\in[C]}\alpha_c\beta_c - \sum_{c\notin S}\alpha_c\beta_c$$

$$\leq \sum_{c\in[C]}\alpha_c\beta_c - \sum_{c\notin S}\dot{\mathbf{w}}_c\tilde{\mathbf{h}}_c$$

$$\leq C\alpha\beta - \sum_{c\notin[C]-S}\dot{\mathbf{w}}_c\tilde{\mathbf{h}}_c$$

$$\leq C\alpha\beta - (1-\delta)\overrightarrow{\mathbf{w}}\cdot\overrightarrow{\mathbf{h}} + (C-1)\sqrt{\frac{2\epsilon_2'\delta(1-\delta)}{\lambda}}$$

Let $\alpha_c = \sqrt{\frac{1}{N}\sum_{i=1}^N\|\mathbf{h}_{c,i}\|^2}$ and $\beta_c = \|\dot{\mathbf{w}}_c\|$. Note that since $-\frac{1}{C-1}\overrightarrow{\mathbf{w}}\cdot\overrightarrow{\mathbf{h}} = -\frac{C}{C-1}\alpha\beta + \epsilon_3'$, there is $\overrightarrow{\mathbf{w}}\cdot\overrightarrow{\mathbf{h}} = C\alpha\beta - (C-1)\epsilon_3'$. Therefore,

$$\sum_{c\in S}\dot{\mathbf{w}}_c\tilde{\mathbf{h}}_c \geq \delta C\alpha\beta - \delta(C-1)\epsilon_3' - (C-1)\sqrt{\frac{2\epsilon_2'\delta(1-\delta)}{\lambda}}$$

$$\sum_{c\in S}\alpha_c\beta_c \leq \delta C\alpha\beta + (1-\delta)(C-1)\epsilon_3' + (C-1)\sqrt{\frac{2\epsilon_2'\delta(1-\delta)}{\lambda}}$$

Therefore, there are at most $\delta C$ classes for which

$$\dot{\mathbf{w}}_c\tilde{\mathbf{h}}_c \leq \alpha\beta - \frac{(C-1)}{C}\epsilon_3' - \frac{C-1}{C}\sqrt{\frac{2\epsilon_2'(1-\delta)}{\delta\lambda}} \tag{25}$$

and also there are at most $\delta C$ classes for which

$$\alpha_c\beta_c \geq \alpha\beta + \frac{(1-\delta)(C-1)}{\delta C}\epsilon_3' + \frac{C-1}{C}\sqrt{\frac{2\epsilon_2'(1-\delta)}{\delta\lambda}} \tag{26}$$

Thus, for at least $(1-2\delta)C$ classes, we have

$$\frac{\dot{\mathbf{w}}_c\tilde{\mathbf{h}}_c}{\alpha_c\beta_c} \geq 1 - \left(\frac{C-1}{C\alpha\beta}\right)\left(\frac{\epsilon_3'}{\delta} - 2\sqrt{\frac{2\epsilon_2'(1-\delta)}{\delta\lambda}}\right) \tag{27}$$

By setting $\epsilon_2' = \epsilon$ and $\epsilon_3' = 0$, we obtain the following upper bound on the cosine of the angle between $\dot{\mathbf{w}}_c$ and $\tilde{\mathbf{h}}_c$:

$$\cos(\angle(\dot{\mathbf{w}}_c, \tilde{\mathbf{h}}_c)) \geq 1 - 2\sqrt{\frac{2\epsilon(1-\delta)}{\delta\lambda}}$$

Using $\lambda = e^{-O(C\alpha\beta)}$, we get the NC3 bound in the theorem:

$$\cos(\angle(\dot{\mathbf{w}}_c, \tilde{\mathbf{h}}_c)) \geq 1 - 2\sqrt{\frac{2\epsilon(1-\delta)e^{O(C\alpha\beta)}}{\delta}} = 1 - O\left(e^{O(C\alpha\beta)}\sqrt{\frac{\epsilon}{\delta}}\right)$$

Let $\mathcal{C}$ denote the set of classes for which the above inequality holds. By applying Lemma B.6 to the set of vectors $\{\mathbf{h}_{c,i}\}$ where $\mathbf{v}_i = \mathbf{h}_{c,i}$, $\mathbf{u} = \frac{\dot{\mathbf{w}}_c}{\|\dot{\mathbf{w}}_c\|}$, and $\beta = 1$, and using lemma B.6 that $intra_c = \|\tilde{\mathbf{h}}_c\|$ we obtain

$$intra_c = \|\tilde{\mathbf{h}}_c\| \geq 1 - 4\left(\frac{C-1}{C\alpha\beta}\right)\left(\frac{\epsilon_3'}{\delta} - 2\sqrt{\frac{2\epsilon_2'(1-\delta)}{\delta\lambda}}\right)$$

for each class $c \in \mathcal{C}$.

Assuming that $\epsilon \ll 1$, then $\epsilon \ll \sqrt{\epsilon}$. Therefore, then worst case bound when $\epsilon \geq \epsilon_2' + \frac{\gamma'}{1+\gamma'}\epsilon_3'$ is achieved when $\epsilon_2' = \epsilon$:

$$intra_c \geq 1 - 8(\frac{C-1}{C\alpha\beta})\sqrt{\frac{2\epsilon(1-\delta)}{\delta\lambda}}$$

Plug in $\lambda = \exp(-O(C\alpha\beta))$ and with simplification we get:

$$intra_c \geq 1 - \frac{(C-1)}{C\alpha\beta}\sqrt{\exp(O(C\alpha\beta))\frac{128\epsilon(1-\delta)}{\delta}} = 1 - O(\frac{e^{O(C\alpha\beta)}}{\alpha\beta}\sqrt{\frac{\epsilon}{\delta}})$$

Now consider the inter-class cosine similarity. Let $m_c = -\frac{C}{C-1}\dot{\mathbf{w}}_c\tilde{\mathbf{h}}_c$, by Lemma B.3 we know that for any set $S$ of $\delta(C-1)$ classes in $[C] - \{c\}$, using the definition that $\dot{\mathbf{w}}_c = \mathbf{w}_c - \tilde{\mathbf{w}}$ there is

$$\sum_{c' \in S}(\dot{\mathbf{w}}_{c'} - \dot{\mathbf{w}}_c)\tilde{\mathbf{h}}_c = \sum_{c' \in S}(\mathbf{w}_{c'} - \mathbf{w}_c)\tilde{\mathbf{h}}_c \leq \delta(C-1)m_c + (C-1)\sqrt{\frac{2\delta\epsilon_{1,c}'}{\exp(m_c)}}$$

Therefore, for at least $(1-\delta)(C-1)$ classes, there is

$$(\dot{\mathbf{w}}_{c'} - \dot{\mathbf{w}}_c)\tilde{\mathbf{h}}_c \leq m_c + \sqrt{\frac{2\epsilon_{1,c}'}{\exp(m_c)\delta}} = -\frac{C}{C-1}\dot{\mathbf{w}}_c\tilde{\mathbf{h}}_c + \sqrt{\frac{2\epsilon_{1,c}'}{\exp(m_c)\delta}} \tag{28}$$

$$\dot{\mathbf{w}}_{c'}\tilde{\mathbf{h}}_c \leq -\frac{1}{C-1}\dot{\mathbf{w}}_c\tilde{\mathbf{h}}_c + \sqrt{\frac{2\epsilon_{1,c}'}{\exp(m_c)\delta}} \tag{29}$$

Combining with equation 25 equation 26, we get that there are at least $(1-2\delta)C \times (1-3\delta)C \geq (1-5\delta)C^2$ pairs of classes $c, c'$ that satisfies the following: for both $c$ and $c'$, equations equation 25 equation 26 are not satisfied (i.e. satisfied in reverse direction), and equation 28 is satisfied for the pair $c', c$. Note that this implies

$$m_c = -\frac{C}{C-1}\dot{\mathbf{w}}_c\tilde{\mathbf{h}}_c \leq -\frac{C}{C-1}\alpha\beta + \epsilon_3' + \sqrt{\frac{2\epsilon_2'(1-\delta)}{\delta\lambda}}$$

and

$$\dot{\mathbf{w}}_{c'}\tilde{\mathbf{h}}_c \leq -\frac{\alpha\beta}{C-1} + \frac{1}{C}(\epsilon_3' + \sqrt{\frac{2\epsilon_2'(1-\delta)}{\delta\lambda}}) + \sqrt{\frac{2\epsilon_{1,c}'}{\exp(m_c)\delta}}$$

We now seek to simplify the above bounds using the constraint that $\epsilon \geq \frac{1}{C}\sum_{c=1}^C \frac{\epsilon_{1,c}'}{C} + \epsilon_2' + \frac{\gamma'}{1+\gamma'}\epsilon_3'$. Note that $\epsilon \ll \sqrt{\epsilon}$, and both $\lambda$ and $\exp(m_c)$ are $\exp(-O(C\alpha\beta))$, therefore, we can achieve the maximum bound by setting $\epsilon_{1,c}' = \epsilon$,

$$\dot{\mathbf{w}}_{c'}\tilde{\mathbf{h}}_c \leq -\frac{\alpha\beta}{C-1} + \exp(O(C\alpha\beta))\sqrt{\frac{2\epsilon}{\delta}}$$

Similarly, we can achieve the smallest bound on $\alpha_c \beta_c$ (the reverse of equation 26)by setting $\epsilon_2' = \epsilon$ and using $\lambda = \exp(-O(\alpha\beta))$ we get for both $c$ and $c'$

$$\alpha_c \beta_c \leq \alpha\beta + \exp(O(\alpha\beta))\sqrt{\frac{2\epsilon}{\delta}}$$

and achieve the largest bound on $\dot{\mathbf{w}}_c \tilde{\mathbf{h}}_c$ (the reverse of equation 25) by setting $\epsilon_2' = \epsilon$ we get for both $c$ and $c'$:

$$\dot{\mathbf{w}}_c \tilde{\mathbf{h}}_c \leq \alpha\beta - \exp(O(C\alpha\beta))\sqrt{\frac{2\epsilon}{\delta}}$$

Therefore, we can apply Lemma B.8 with $\alpha = \alpha_c$, $\beta = \beta_c$, $\epsilon' = \alpha_c\beta_c - \dot{\mathbf{w}}_c\tilde{\mathbf{h}}_c \leq 2\exp(O(C\alpha\beta))\sqrt{\frac{2\epsilon}{\delta}}$ bound to get:

$$\cos_\angle(\dot{\mathbf{w}}_{c'}, \tilde{\tilde{\mathbf{h}}}_c) \leq -\frac{1}{C-1} + \frac{C}{C-1}\frac{\exp(O(C\alpha\beta))}{\alpha\beta}\sqrt{\frac{2\epsilon}{\delta}} + 4(\frac{2\exp(O(C\alpha\beta))}{\alpha\beta}\sqrt{\frac{2\epsilon}{\delta}})^{1/3}$$

$$\leq -\frac{1}{C-1} + O(\frac{e^{O(C\alpha\beta)}}{\alpha\beta}(\frac{\epsilon}{\delta})^{1/6})$$

Where the last inequality is because $\frac{e^{O(C\alpha\beta)}}{\alpha\beta} > 1, \frac{\epsilon}{\delta} < 1$. Finally, we derive an upper bound on $\cos_\angle(\tilde{\tilde{\mathbf{h}}}_{c'}, \tilde{\mathbf{h}}_c)$ and thus intra-class cosine similarity by combining the above bounds. Note that for $\frac{\pi}{2} < a < \pi$ and $0 < b < \frac{pi}{2}$ we have:

$$\cos(a - b) = \cos(a)\cos(b) + \sin(a)\sin(b)$$
$$\leq \cos(a) + \sin(b)$$
$$\leq \cos(a) + \sqrt{1 - \cos^2(b)}$$
$$\leq cos(a) + \sqrt{2(1 - \cos(b))}$$

by equation 27 we get that

$$\cos_\angle(\dot{\mathbf{w}}_{c'}, \tilde{\tilde{\mathbf{h}}}_{c'}) \geq 1 - (\frac{C-1}{C\alpha\beta})(\frac{\epsilon_3'}{\delta} - 2\sqrt{\frac{2\epsilon_2'(1-\delta)}{\delta\lambda}}) \geq 1 - \frac{\exp(O(C\alpha\beta))}{\alpha\beta}\sqrt{\frac{2\epsilon}{\delta}}$$

Therefore,

$$\cos_\angle(\tilde{\tilde{\mathbf{h}}}_{c'}, \tilde{\mathbf{h}}_c) \leq \cos_\angle(\dot{\mathbf{w}}_{c'}, \tilde{\tilde{\mathbf{h}}}_c) + \sqrt{2(1 - \cos_\angle(\dot{\mathbf{w}}_{c'}, \tilde{\tilde{\mathbf{h}}}_{c'}))}$$

$$\leq -\frac{1}{C-1} + \frac{C}{C-1}\frac{\exp(O(C\alpha\beta))}{\alpha\beta}\sqrt{\frac{2\epsilon}{\delta}} + 4(\frac{2\exp(O(C\alpha\beta))}{\alpha\beta}\sqrt{\frac{2\epsilon}{\delta}})^{1/3} + \sqrt{\frac{\exp(O(C\alpha\beta))}{\alpha\beta}\sqrt{\frac{2\epsilon}{\delta}}}$$

$$= -\frac{1}{C-1} + O(\frac{e^{O(C\alpha\beta)}}{\alpha\beta}(\frac{\epsilon}{\delta})^{1/6})$$

Since $\|\tilde{\tilde{\mathbf{h}}}_c\| \leq 1$, there is

$$\tilde{\tilde{\mathbf{h}}}_{c'} \cdot \tilde{\tilde{\mathbf{h}}}_c = \|\tilde{\tilde{\mathbf{h}}}_{c'}\|\|\tilde{\tilde{\mathbf{h}}}_c\|\cos_\angle(\tilde{\tilde{\mathbf{h}}}_{c'}, \tilde{\tilde{\mathbf{h}}}_c) \leq -\frac{1}{C-1} + O(\frac{e^{O(C\alpha\beta)}}{\alpha\beta}(\frac{\epsilon}{\delta})^{1/6})$$

Applying B.5 shows the bound on inter-class cosine similarity. Note that although this bound holds only for $1 - 5\delta$ fraction of pairs of classes, changing the fraction to $1 - \delta$ only changes $\delta$ by a constant factor and does not affect the asymptotic bound. $\qquad\square$

### B.3    Proof of Theorem 2.2

**Theorem B.2** (Detailed Version of 2.2). *For an neural network classifier without bias terms trained on a dataset with the number of classes $C \geq 3$ and samples per class $N \geq 1$, under the following assumptions:*

1. *The network contains an batch normalization layer without bias term before the final layer with trainable weight vector $\boldsymbol{\gamma}$;*

2. *The layer-peeled regularized cross-entropy loss with weight decay $\lambda < \frac{1}{\sqrt{C}}$*

$$\mathcal{L}_{\text{reg}} = \frac{1}{CN} \sum_{c=1}^{C} \sum_{i=1}^{N} \mathcal{L}_{\text{CE}} \left( f(\boldsymbol{x}_{c,i}; \boldsymbol{\theta}), \boldsymbol{y}_c \right) + \frac{\lambda}{2} (\|\boldsymbol{\gamma}\|^2 + \|\mathbf{W}\|_F^2)$$

*satisfies $\mathcal{L}_{\text{reg}} \leq m_{\text{reg}} + \epsilon$ for small $\epsilon$; where $m_{reg}$ is the minimum achievable regularized loss*

*then for at least $1 - \delta$ fraction of all classes , with $\frac{\epsilon}{\delta} \ll 1$, $\epsilon < \lambda$ and for small constant $\kappa > 0$ and $\rho = (\frac{Ce}{\lambda})^{\kappa C}$ there is*

$$intra_c \geq 1 - \frac{C-1}{C} \sqrt{\frac{128\rho\epsilon(1-\delta)}{\delta}} = 1 - O\left( \left(\frac{C}{\lambda}\right)^{O(C)} \sqrt{\frac{\epsilon}{\delta}} \right),$$

*and also for a cosine similarity representation of NC3 in Papyan et al. (2020):*

$$\cos_{\angle}(\dot{\mathbf{w}}_c, \tilde{\mathbf{h}_c}) \geq 1 - 2\sqrt{\frac{2\rho\epsilon(1-\delta)}{\delta}} = 1 - O\left( \left(\frac{C}{\lambda}\right)^{O(C)} \sqrt{\frac{\epsilon}{\delta}} \right),$$

*and for at least $1 - \delta$ fraction of all pairs of classes $c, c'$, with $\frac{\epsilon}{\delta} \ll 1$, there is*

$$inter_{c,c'} \leq -\frac{1}{C-1} + \frac{C\rho}{C-1}\sqrt{\frac{2\epsilon}{\delta}} + 4(\rho\sqrt{\frac{2\epsilon}{\delta}})^{1/3} + \sqrt{\rho\sqrt{\frac{2\epsilon}{\delta}}} = -\frac{1}{C-1} + O(\left(\frac{C}{\lambda}\right)^{O(C)} (\frac{\epsilon}{\delta})^{1/6})$$

*Proof.* Let $\boldsymbol{\gamma}^*$ and $\boldsymbol{W}^*$ be the weight vector and weight matrix that achieves the minimum achievable regularized loss. Let $\alpha = \|\boldsymbol{\gamma}\|$ and $\beta = \frac{\|\boldsymbol{W}\|_F}{\sqrt{C}}$, and $\alpha^*$ and $\beta^*$ represent the values at minimum loss accordingly. According to Proposition B.4, we know that $\sqrt{\frac{1}{N} \sum_{i=1}^{N} \|\mathbf{h}_i\|_2^2} = \|\boldsymbol{\gamma}\|_2 = \alpha$. From Theorem B.1 we know that, under fixed $\alpha\beta$, the minimum achievable unregularized loss is $\log(1 + (C-1)\exp(-\frac{C}{C-1}\alpha\beta))$. Since only the product $\gamma = \alpha\beta$ is of interest to Theorem B.1, we make the following observation:

$$\mathcal{L}_{\text{reg}} = \frac{1}{CN} \sum_{c=1}^{C} \sum_{i=1}^{N} \mathcal{L}_{\text{CE}} \left( f(\boldsymbol{x}_{c,i}; \boldsymbol{\theta}), \boldsymbol{y}_c \right) + \frac{\lambda}{2} (\|\boldsymbol{\gamma}\|^2 + \|\mathbf{W}\|_F^2)$$

$$\geq \log(1 + (C-1)\exp(-\frac{C}{C-1}\alpha\beta)) + \frac{\lambda}{2}(\alpha^2 + C\beta^2)$$

$$\geq \log(1 + (C-1)\exp(-\frac{C}{C-1}\gamma)) + \sqrt{C}\lambda\gamma$$

$$\geq \min_{\gamma} \log(1 + (C-1)\exp(-\frac{C}{C-1}\gamma)) + \sqrt{C}\lambda\gamma$$

Now we analyze the properties of this function. For simplicity, we combine $\sqrt{C}\lambda$ into $\lambda$ in the following proposition:

**Proposition B.1.** *The function $f_\lambda(\gamma) = \log\left(1 + (C-1)\exp(-\frac{C}{C-1}\gamma)\right) + \lambda\gamma$ have minimum value*

$$f_\lambda(\gamma^*) = \log(1 - \frac{C-1}{C}\lambda) + \frac{C-1}{C}\lambda \log\left(\frac{C - (C-1)\lambda}{\lambda}\right)$$

*achieved at $\gamma^* = O(\log(\frac{1}{\lambda}))$ for $\lambda < 1$. Furthermore, for any $\gamma$ such that $f_\lambda(\gamma) - f_\lambda(\gamma^*) \leq \epsilon \ll \lambda$, there is $|\gamma - \gamma^*| \leq \sqrt{O(1/\lambda)\epsilon}$*

*Proof.* Consider the optimum of the function by setting the derivative to 0:

$$g'_\lambda(\gamma^*) = -\frac{C}{C-1}\frac{(C-1)\exp(-\frac{C}{C-1}\gamma^*)}{\left(1+(C-1)\exp(-\frac{C}{C-1}\gamma^*)\right)} + \lambda = 0$$

$$\frac{C-1}{C}\lambda = 1 - \frac{1}{1+(C-1)\exp(-\frac{C}{C-1}\gamma^*)}$$

$$1+(C-1)\exp(-\frac{C}{C-1}\gamma^*) = \frac{1}{1-\frac{C-1}{C}\lambda}$$

$$\gamma^* = \frac{C-1}{C}\log\left(\frac{C-(C-1)\lambda}{\lambda}\right) < \log(\frac{C}{\lambda})$$

Plugging in $\gamma^* = \frac{C-1}{C}\log\left(\frac{C-(C-1)\lambda}{\lambda}\right)$ to the original formula we get:

$$f_\lambda(\gamma^*) = \log(1-\frac{C-1}{C}\lambda) + \frac{C-1}{C}\lambda\log\left(\frac{C-(C-1)\lambda}{\lambda}\right)$$

Note that since $\gamma \geq 0$, the optimum point is only positive when $\lambda \leq 1$.

Now consider the case where the loss is near-optimal and $\gamma = \gamma^* + \epsilon'$ for $\epsilon' \ll 1$:

$$\log\left(1+(C-1)\exp(-\frac{C}{C-1}(\gamma^*+\epsilon'))\right) + \lambda(\gamma^*+\epsilon')$$

$$\geq \log\left(1+(C-1)\exp(-\frac{C}{C-1}\gamma^*)(1-\frac{C}{C-1}\epsilon'+\frac{\epsilon'^2}{2})\right) + \lambda(\gamma^*+\epsilon')$$

$$\geq \log\left(1+(C-1)\exp(-\frac{C}{C-1}\gamma^*)\right) + \frac{(C-1)\exp(-\frac{C}{C-1}\gamma^*)}{\left(1+(C-1)\exp(-\frac{C}{C-1}\gamma^*)\right)}(-\frac{C}{C-1}\epsilon'+\frac{\epsilon^2}{2}) + \lambda(\gamma^*+\epsilon')$$

By definition of $\gamma^*$ as the optimal $\gamma$, the first-order term w.r.t. $\epsilon'$ must cancel out. Also, by plugging in $\gamma^*$, the coefficient of $\frac{\epsilon'^2}{2}$ is $\frac{C-1}{C}\gamma$. Therefore,

$$\log\left(1+(C-1)\exp(-\frac{C}{C-1}(\gamma^*+\epsilon'))\right) + \lambda(\gamma^*+\epsilon')$$

$$\leq \log\left(1+(C-1)\exp(-\frac{C}{C-1}\gamma^*)\right) + \lambda\gamma^* + \frac{C-1}{C}\lambda\epsilon'^2$$

Conversely, for any $\epsilon \ll 1$ for which $g(\gamma) \leq g(\gamma^*)+\epsilon$, there must be $|\gamma-\gamma^*| \leq \sqrt{\frac{C\epsilon}{(C-1)\lambda}}$ $\qquad\square$

Thus, the minimum achievable value of the regularized loss is

$$m_{\text{reg}} = \log(1-\frac{C-1}{\sqrt{C}}\lambda) + \frac{C-1}{\sqrt{C}}\lambda\log\left(\frac{\sqrt{C}}{\lambda}-(C-1)\right)$$

Now, consider any $\mathbf{W}$ and $\boldsymbol{\gamma}$ that achieves near-optimal regularized loss $\mathcal{L}_{\text{reg}} = m_{\text{reg}}+\epsilon$ for very small $\epsilon$. Recall that $\alpha = \|\boldsymbol{\gamma}\|$, $\beta = \frac{\|\mathbf{W}\|_F}{\sqrt{C}}$, $\gamma = \alpha\beta$. According to Proposition B.1 we know that $|\gamma-\gamma^*| \leq \sqrt{\frac{C\epsilon}{(C-1)\lambda}}$. Therefore, $\gamma \leq \gamma^* + \sqrt{\frac{C\epsilon}{(C-1)\lambda}} = O(\log(C/\lambda)) + \sqrt{\frac{C\epsilon}{(C-1)\lambda}}$. Also, note that $\mathcal{L}_{\text{reg}} - f_{\sqrt{C}\lambda}(\gamma) \leq \mathcal{L}_{\text{reg}} - f_{\sqrt{C}\lambda}(\gamma^*) = \epsilon$, where $f_{\sqrt{C}\lambda}(\gamma)$ is the minimum unregularized loss according to Theorem B.1. Therefore, we can apply Theorem B.1 with $\alpha\beta = \gamma < O(\log(C/\lambda)) + \sqrt{\frac{C\epsilon}{(C-1)\lambda}}$ and the same $\epsilon$ to get the results in the theorem.

$$\square$$

