# OpenReview forum: "Towards Understanding Neural Collapse: The Effects of Batch Normalization and Weight Decay"
_TMLR — Accepted by TMLR_

### Review · Reviewer_eHk8 · 2026-02-19

**Summary Of Contributions:**

This paper presents a theoretical investigation of the Neural Collapse (NC)  phenomenon in unconstrained
feature models in the presence of weight decay (WD) and batch normalization (BN). In particular, in the near-optimal loss regime. The authors also present experimental evidence to demonstrate the role of WD and BN in achieving collapse on real and synthetic datasets.

**Audience:**

Yes

**Audience Explanation:**

Experimentally, this paper does not offer new insights about NC or the role of WD/BN. There have been studies on it already. From a theoretical perspective, the UFM analysis in the near optimal-loss regime might be of interest to some readers. However, UFM itself has its own limitations since it does not account for the underlying data distribution. Overall, I am not confident about significant interest.

**Claims And Evidence:**

No

**Claims Explanation:**

The paper aims to answer two questions (as per section 1): a minimal set of conditions to guarantee NC emergence, and new insights on the use of WD and BN.


The derivation of near-optimal proximity bounds for cosine-similarity-based Neural Collapse (NC) under batch normalization (BN) and weight decay (WD) is careful and rigorous. Also, the first question implies that there are potentially many conditions that confound the emergence of NC, and this work presents a minimal set of conditions for NC emergence. However, the work assumes BN and WD to be present and states probabilistic results about NC for some classes (Theorem 2.2). The authors might want to reconsider the claim here.

Furthermore, the result is an extension of the work by [1] to the analysis of NC in the near-optimal loss regime. As previous papers have studied NC with WD and BN, the experimental insights presented in this paper need to be carefully positioned to showcase the "newness" of the results. The main concern is that UFMs do not fully capture the training dynamics and are only meant for analyzing the global optima of classifiers. So, even the near-optimal loss regime does not seem to provide new experimental insights. It would be great if the authors could clarify this aspect.


[1] Jianfeng Lu and Stefan Steinerberger. Neural collapse under cross-entropy loss. Applied and Computational Harmonic Analysis, 59:224–241, 2022. ISSN 1063-5203. Special Issue on Harmonic Analysis and Machine Learning.

[2] Súkeník, Peter, Christoph Lampert, and Marco Mondelli. "Neural collapse vs. low-rank bias: Is deep neural collapse really optimal?." Advances in Neural Information Processing Systems 37 (2024): 138250-138288.

[3] Dang, Hien, et al. "Neural collapse for cross-entropy class-imbalanced learning with unconstrained ReLU features model." Proceedings of the 41st International Conference on Machine Learning. 2024.

**Requested Changes:**

1. In Section 2.1, the setup describes a bias-free formalization, but the equation below includes a bias term.
2. Mixing the loss and NC values in Figure 3 is not appropriate in my opinion since the y-axis measures two entirely different metrics.
3. Please adjust the spacing of figures for a tighter text placement. For ex: page 8.

---

> ### Author Response · Authors · 2026-03-10
> **Response to Reviewer eHk8**
>
> Thank you for the careful review and constructive suggestions. We agree that the main value of this work should be communicated precisely, and we have revised the manuscript accordingly. Below, we respond point by point.
>
>
> **Scope of the theoretical results**
>
> We agree that our results should not be framed as "minimal conditions" or as establishing a causal mechanism that BN/WD cause NC during training dynamics. Our theoretical contribution is a perturbation analysis: it quantifies how close a solution must be to the NC geometry, conditional on being near-optimal and under WD and BN. We have revised the language throughout the manuscript to clarify that the results are conditional perturbation bounds rather than a characterization of all possible mechanisms leading to NC.
>
>
> **Novelty over Lu and Steinerberger.**
>
> We appreciate the reviewer's point and agree that the relationship to Lu and Steinerberger (2022) should be clarified more explicitly.
>
> Lu and Steinerberger (2022) prove that the global minimizer of the CE loss under a unit-norm constraint on features corresponds to the simplex ETF configuration. This is an exact optimality result under a hard norm constraint ($|\mathbf{h}|=1$).
>
> Our work differs in three important aspects:
>
> 1. **Near-optimal regime vs. exact optimality.**
>
> Prior results characterize the geometry of the exact optimum. Our Theorem 2.2 instead provides quantitative bounds in the near-optimal regime, showing how deviations in NC metrics depend on the optimality gap $\epsilon$. This provides a perturbation analysis of NC geometry when the loss is close to optimal.
>
>
> 2. **Soft regularization vs. hard constraints.** Lu and Steinerberger (2022) assume features lie on the unit sphere. In contrast, we analyze the more realistic setting where feature norms are controlled through $L_2$ regularization (weight decay) rather than hard constraints. This leads to a different analytical framework.
>
>
> 3. **Role of WD strength.** Our bounds explicitly characterize how the weight decay parameter $\lambda$ influences the tightness of NC guarantees. Larger $\lambda$ leads to stronger norm control and therefore tighter bounds on NC deviations. This quantitative dependence does not appear in prior optimality results.
>
> We have revised the paper to make this comparison explicit and position our contribution more precisely as a "perturbation analysis."
>
> **UFM doesn't capture training dynamics.**
>
> We agree that UFMs do not capture the detailed training dynamics of deep neural networks. For this reason, our results are framed as *optimization-agnostic*: the bounds apply to any solution that achieves near-optimal loss under the assumed norm control, regardless of how that solution is obtained.
>
> Moreover, our analysis focuses on the near-optimal regime, which is more realistic in practice than exact optimality. This allows us to establish a quantitative relationship between loss optimality and proximity to the NC geometry, which can be directly examined in empirical training trajectories.
>
> **Minor.**
> 1. The equation in Section 2.1 originally contained a bias term that was not used in any of the proofs. We have corrected this typo — the equation now accurately reflects the bias-free setting.
>
> 2. We have revised Figure 3 to use dual y-axes: the left y-axis shows NC metrics (cosine similarity, solid and dashed lines) and the right y-axis shows training loss on a log scale (dotted lines).
>
> 3. We have tightened figure spacing throughout multiple figures of the paper.

---

> > ### Comment · Reviewer_eHk8 · 2026-03-24
> >
> > Thank you for the responses.

---

### Review · Reviewer_oHQt · 2026-02-20

**Summary Of Contributions:**

The paper studies the optimality and near-optimality conditions on the strength of neural collapse in the unconstrained features model (UFM) with the cross-entropy loss, in the presence of weight decay and regularized batch norm. The theoretical contribution is the establishment of quantitative near-optimal upper-bounds on how much can a solution of the UFM deviate from perfect collapse if it is near-optimal. This deviation is a function of the optimality gap, but also the regularization strength (norm of the features) - the stronger the regularization (the smaller the features), the closer the features and weights must be to a perfect NC configuration (to still fall within the optimality gap). On the experimental side, the authors demonstrate the importance of batch norm in establishing strong collapse and also how its presence changes the effect of weight decay -- if the BN is present, WD has a more-or-less monotonic effect on the emergence of NC (the more the better), while if the BN is absent, WD has fairly unpredictable (but overall a bit smaller) effect on the strength of the NC in the converged solutions. This is supplemented with results that look at the evolution of the NC metrics across the training, where the BN guarantees more correlation of the training loss with the metrics and overall more monotonicity, while its absence usually means quite quick stagnation of the metrics, well before convergence. Finally, authors provide anti-correlation plots between the norm of the features and the strength of the NC metrics.

Here I briefly outline the key strengths and weaknesses, on which I elaborate in the next fields:

**Strengths:**

- To me the main strength is the empirical demonstration of the significance of the batch norm (and because it is regularized, one could say it is instead the presence of explicit feature regularization) on the emergence of NC and the role of WD.

- Besides, computing good lower-bounds on the local strong convexity of the cross-entropy loss and utilizing these correctly for downstream properties (such as the cosine similarities) are tools that are interesting in context, but could potentially be used also in other contexts.

**Weaknesses:**

- The main weakness is the misinterpretation of the theory results by claiming that they *imply* or *suggest* stronger NC guarantees with stronger WD and in the presence of BN. This misinterprets the causal direction of the theorems.

- There is quite poor connection between the theory and the experiments.

- In some cases, the experiments are missing convergence ablations.

- The related work discussion is modest and outdated. It covers 2 papers from 2023 and 0 papers from 2024+, despite the speed at which the NC theory/practice has been evolving.

**Additional Comments:**

I have a couple of questions and typo catches for the authors.

- 1. In Figure 3 left, it seems the average inter-class cosine similarity is reaching an absolute lower bound and yet the max still deviates from it. This seems to be mathematically impossible. Can you explain that?

- 2. In Figure 5 the worst-case inter-class cosine similarities are often reaching sky-high values (basically pure 1), yet the loss converges to low values (I suppose). How is this possible?

- 3. Why did you omit the NC3 property from your experiments?

- 4. Can you explain the following quote from your paper: *"We perform this experiment only on synthetic data due to the existence of multiple BN layers in real-world
models such as VGG, which makes such operations ambiguous."* I don't fully understand this argument. Can you please elaborate on that?

I also have a few typos or formatting suggestions:

- The first sentence of section 3.4 should probably say "smaller feature norm" instead of "higher".

- Please plot theoretical lower-bounds on the inter-class cosine similarity as a horizontal line in all the relevant plots.

- Please fix the small parentheses "()" in big formulas, such as in the last big formula before section 2.4, but also in the appendix.

**Audience:**

Yes

**Audience Explanation:**

I am confident that *both* the quantitative loss premium - NC metrics relationship *and* the experiments showing that the explicit feature norm regularization and optimization breaks the issues related with MLP with WD loss landscape and enable NC (even get stronger as the WD goes up) are interesting findings that not only NC theorists would benefit from.

**Broader Impact Concerns:**

I do not have any concerns about the broader impact.

**Claims And Evidence:**

No

**Claims Explanation:**

I am not challenging the technical accuracy of the theory and experimental results. Instead, I am challenging some interpretations that are core to the paper's contribution.

- 1. As mentioned above, the biggest issue is the misinterpretation and thereby oversell of the theoretical results. As the authors write: *"The quantitative bounds of our theorem imply that smaller last-layer feature and weight norms can provide
stronger guarantees on NC."* Unfortunately this statement is conditional on the loss being close to its optimal value. There is no evidence presented in the theory that would suggest that weight decay and batch norm *cause* better NC metrics by, for instance, converging faster or closer to the theoretical minimum. The only thing the theorems really say is that the softmax loss is more sensitive to perturbations when the scores are smaller in their absolute values, simply due to the local strong convexity constant increasing. Therefore, the correct interpretation of the theory is: BN and larger values of WD make features smaller, which in turn makes the loss *more sensitive* to deviations, which in turn *only allows* stricter NC measures to reach that loss premium. Therefore, the theorems do not establish any causal effect between the use of BN and WD and the resulting NC strength. On top of that, the theory also doesn't present a counterfactual -- what would happen, if the BN was not present. In fact, the vast literature on UFM with CE loss has taught us that NC is optimal/converged to in basically any reasonable setting (which is different from MSE loss where some settings do not enjoy exact convergence).

- 2. Staying with the theory, the authors use the BN and argue that it is the BN that is responsible for most of the qualitative behaviors (of course supplemented by the WD). However, this is also a misinterpretation. The authors use $l_2$ regularization on the *learnable* BN parameters. When the parameters of the BN are learnable and regularized, I would not consider it BN anymore, but rather direct feature $l_2$ regularization and optimization. Therefore, the evidence presented in the paper proves that adding a direct feature $l_2$ regularization is the main reason why the NC gets better. This is intuitively correct, because adding an explicit feature regularizer makes the loss landscape closer to the ``pure"" UFM setting. Therefore, using regularized BN effectively just bridges the gap between the ideal UFM scenario and the dirty end-to-end loss landscape.

- 3. The authors claim their experiments to support their theoretical claims. However, there is quite little connection there. In particular a lot of the experimental results revolve around comparing BN and no-BN settings, which is interesting, but because the theory doesn't study the no-BN counterfactual, this is simply an independent piece of research. The counterfactual is studied in related work, where BN is often omitted from the UFM setup, and the optimality of NC is reached in those works too, therefore what we see in experiments with no-BN must happen for a reason which is not even explainable by the UFM model. The experiments do support that in the presence of BN, increasing the WD leads to monotonic improvements in the NC metrics. However, it is not clear whether this happens for the same reason as the theory would suggest, or because the stronger WD simply speeds up and improves the convergence overall.

- 4. The experiments that show a negative correlation between a fixed-norm LN scale parameter and the NC metrics miss an ablation on the speed of convergence. It is absolutely not clear, whether the results that we see aren't just a consequence of the fact that smaller features lead to faster convergence.

**Requested Changes:**

Changes that are necessary for my recommendation of acceptance (in the light of the previous fields):

- Significantly restructure the claims, interpretations, messaging and narratives of the paper. In particular, the interpretation of the theoretical results should not reach beyond perturbation analysis of the NC metrics against loss premium and the explanation of how this relates to the local strong convexity and the other tools used. The experiments should be presented as more of self-standing claims that *also* concern BN and WD, but not as a direct support of the theory (some exceptions are possible if the ablations mentioned in the previous fields are fixed). The use of BN should be rephrased as direct feature norm regularization and optimization, because that is what the BN really does if we allow the scale parameters to be learnable and regularized.

- Perform the ablations on the experiments mentioned in the previous fields.

- Significantly extend the related work discussion, including papers from 2023+.

Changes that would strengthen the paper but are not necessary for a publication at TMLR:

- The authors should discuss possible reasons why WD fails to deliver NC in the BN-free setting. For instance, it has been demonstrated that vanilla MLPs regularized with WD suffer from low-rank bias that can break some of the NC properties [1]. This could be a plausible explanation why WD is of no help in these settings.

[1] Súkeník, Peter, Christoph Lampert, and Marco Mondelli. "Neural collapse vs. low-rank bias: Is deep neural collapse really optimal?." Advances in Neural Information Processing Systems 37 (2024): 138250-138288.

---

> ### Author Response · Authors · 2026-03-10
> **Response for Reviewer oHQt**
>
> We thank the reviewer for the careful reading and constructive feedback, which helped us significantly clarify the scope and interpretation of our results. Below, we address the reviewer’s concerns in detail.
>
>
> **Theory interpretation / overstated claims:**
> We agree with the reviewer that the theoretical results should not be interpreted as establishing a causal mechanism by which WD or BN causes neural collapse during training.
>
> Our theoretical contribution is a perturbation analysis. Specifically, Theorem 2.2 provides quantitative bounds on how far a near-optimal solution of the regularized UFM objective can deviate from the NC geometry. The bounds show that the deviation scales with the optimality gap and norm-control parameters.
>
> In particular, the result shows that under norm control (induced by weight decay and regularized BN), smaller feature norms tighten the admissible deviation from NC for solutions with a fixed loss gap. This effect can be understood through the local strong convexity of the cross-entropy loss when logits have a smaller magnitude.
>
>
> Following your suggestion, we revised the manuscript to make this perturbation structure explicit. In particular, we clarified in the abstract, introduction, and Section 2 that the results are conditional bounds: when the loss is near-optimal, stronger norm control leads to tighter guarantees on the proximity to the NC configuration.
>
> We also refined the language throughout the paper to clearly distinguish the theoretical statement (perturbation bounds conditional on loss and norms) from empirical observations about training dynamics.
>
> **Weak connection between theory and experiments:**
> We agree that the theory does not analyze counterfactual training settings, such as the absence of BN. Our intention was not to claim that the experiments directly verify the theoretical mechanism. Instead, we view the theoretical and experimental results as complementary:
> - The theory characterizes the asymptotic constraints and dependency of near-optimal solutions under norm control.
> - The experiments investigate how practical training mechanisms (BN and WD) influence the emergence and strength of NC in real networks, guided by the qualitative insights from the theory.
>
> In particular, the experiments show that when BN-based norm control is present, NC metrics evolve more monotonically and correlate more strongly with training loss, whereas in the absence of BN the NC metrics often stagnate early.
>
> Following the reviewer’s suggestion, we clarified this distinction in the revised manuscript. The theoretical analysis establishes perturbation bounds under norm control, while the experiments explore how training mechanisms such as BN and WD influence the emergence of NC in practice. These two components are intended to provide complementary perspectives rather than a direct experimental verification of the theory.
>
> While prior work has documented the presence of NC in various settings, our experiments specifically examine how BN-based norm control interacts with WD to shape the *trajectory and monotonicity* of NC emergence during training, which has not been systematically studied in previous work.
>
>
> **BN interpretation:**
> We appreciate the reviewer’s point that when BN scale parameters are learnable and regularized, the mechanism effectively acts as explicit feature norm regularization. We agree with this interpretation and have revised the manuscript to clarify this point.
>
> As clarified in Section 2.1, in our formalization, the regularized BN layer effectively acts as an explicit feature norm regularizer: BN first normalizes features to zero mean and unit variance, and the $L_2$-regularized scale parameter $\gamma$ directly controls the norm of the resulting features.
>
> In this sense, the theoretical bounds depend on the presence of norm control rather than on the specific architectural implementation of BN.

---

> ### Author Response · Authors · 2026-03-10
> **Response for Reviewer oHQt (part 2)**
>
> **Convergence Ablation:**
>
> We agree that optimization speed could confound the interpretation of the NC–norm relationship. In particular, smaller feature norms might in principle make optimization easier, which could make NC appear stronger simply because the model converges faster. We address this concern as follows.
>
>
> 1. **NC evolution during training.** Figures 3 and 8 show the NC metrics evolution as a function of training time. These figures show that with BN, the NC metrics improve steadily as training proceeds. In contrast, in models without BN, the NC metrics quickly stagnate even though the training loss continues to decrease substantially. This indicates that reducing loss alone does not automatically lead to stronger NC, underscoring the role of BN-based feature-norm control.
>
> 2.**NC as a function of training loss.** To further remove the effect of training speed, Figure 10 plots NC metrics directly against the training loss by sampling checkpoints during training. This allows comparison of models at comparable loss levels rather than comparable training times. The plots show that, at similar loss levels, BN models consistently exhibit stronger NC than no-BN models, indicating that the NC improvement cannot be explained solely by faster convergence.
>
> 3. **Fixed-$|\gamma|$ convergence ablation.** For the fixed-$|\gamma|$ experiments in Section 3.4, we additionally report the final training loss together with the NC metrics in Figure 4. Across 72 runs (6 $\gamma$ values × 2 depths × 2 datasets × 3 seeds), smaller $|\gamma|$ produces stronger NC while achieving slightly higher final training loss. This further confirms that the stronger NC observed under tighter norm control is not a byproduct of faster optimization or better data fitting.
>
>
>
>
> **Discuss why WD fails without BN (low-rank bias).**
>
>
> Thank you for the reference. We agree that our current theory and experiments do not provide a direct explanation for why WD fails to induce strong NC in the BN-free setting, and this remains a limitation of our work. We have therefore added a discussion in the Related Work section (Section 1.2) citing Súkeník et al. (NeurIPS 2024), who provide a plausible explanation through the low-rank bias induced by multi-layer $L_2$ regularization.
>
> In particular, Súkeník et al. show that weight decay across layers can favor low-rank intermediate representations, which may compete with the geometry required for deep neural collapse. Intuitively, if the learned representation is biased toward a low-dimensional subspace, it may fail to realize the full simplex-like structure needed for strong NC. This provides a plausible explanation for our empirical observation that WD alone, without BN-based norm control, does not reliably promote NC.
>
>
> **Related Work**
> We have also expanded the related work section to include several recent works on neural collapse and related optimization phenomena, to better position our contributions within the rapidly evolving NC literature.
>
>
> **Figure 3 --- avg vs max inter-class cosine similarity.**
>
> We respectfully clarify that this behavior is **not** mathematically impossible.
>
> The quantity $-1/(C-1)$ is a lower bound on the **average** pairwise inter-class cosine similarity (from $\|\sum_c \hat{m}_c\|^2 \geq 0$). Critically:
>
> 1. **Individual pairwise cosines can go below $-1/(C-1)$.** The bound applies only to the average, not to each pair.
>
> 2. **Average at $-1/(C-1)$ does not imply NC.** It only implies $\sum_c \hat{m}_c = 0$ (the class-mean directions are centered), which is a much weaker condition than the simplex ETF. For example, with $C=4$, consider two pairs of nearly-antipodal vectors in orthogonal subspaces: $\hat{m}_{1,2} = (1/2, \pm\sqrt{3}/2, 0, \ldots)$, $\hat{m}_{3,4} = (-1/2, 0, \pm\sqrt{3}/2, \ldots)$. The average inter-class cosine is **exactly** $-1/3$ (optimal), yet the max is $-1/4$ --- a gap of $0.083$. This configuration is clearly not a simplex ETF.
>
> 3. **Only the worst-case (max) reaching $-1/(C-1)$ guarantees simplex ETF.** When $\text{max} = \text{avg} = -1/(C-1)$, all pairs must be equal, which is the simplex ETF.
>
> The gap between average and worst-case inter-class cosine visible in our figures is therefore an informative feature, not a bug. To make this clearer, we have added horizontal reference lines at $-1/(C-1)$ to all inter-class plots and added a caveat in the caption explaining that reaching this average alone does not imply NC2.

---

> ### Author Response · Authors · 2026-03-10
> **Response for Reviewer oHQr (part 3)**
>
> **Figure 5 (now Figure 6) --- worst-case cosine near 1 but loss low.**
>
> Thank you for pointing this out. The key point is that the plotted quantity is the **worst-case inter-class cosine** across all class pairs. With $C=10$ classes, there are $\binom{10}{2}=45$ pairs, so a value close to 1 only indicates that one or a few pairs of class means become nearly collinear.
>
> This does not imply that the overall feature geometry is degenerate. Indeed, as shown in the corresponding **average inter-class cosine** plots, the average values remain close to the simplex ETF value. Thus, the high worst-case cosine reflects a small number of problematic pairs rather than the global class geometry.
>
>
> **Why omit NC3 from experiments?**
>
> NC3 (self-duality: $\cos(\mathbf{w}_c, \tilde{\mathbf{h}}_c) \to 1$) is included in our theoretical results (Theorems 2.1 and 2.2). In the original experimental figures, we focused on NC1 and NC2, which depend only on the feature geometry and can be directly compared within the same plots. This allowed us to present the main geometric trends more clearly.
>
> Following the reviewer’s suggestion, we have now added NC3 self-duality plots (Figure 5). The results show the same qualitative behavior as NC1 and NC2: BN models converge to stronger self-duality faster and more reliably than models without BN.
>
>
>
> **Clarify "multiple BN layers" argument.**
>
> We have revised Section 3.4 to read: *"We perform this experiment only on synthetic data (single BN layer before the final linear layer) because real-world architectures such as VGG contain multiple BN layers throughout the network, and fixing the scale parameter of only the last BN layer while others remain trainable does not cleanly isolate the effect predicted by our theory."*
>
> **Minor.**
> 1. Thank you for pointing this out. This was a typo, and the sentence has been corrected to “smaller feature norm” in the revised version.
>
> 2. We have added horizontal dashed gray lines at $-1/(C-1)$ (the simplex ETF inter-class cosine similarity) to the inter-class cosine similarity plots.
>
> 3. We have replaced improperly sized parentheses with `\left(` `\right)` throughout the theorem statements, lemmas, and proofs in both the main text and appendix.

---

> > ### Comment · Reviewer_oHQt · 2026-03-13
> > **Thank you for your answers**
> >
> > Thank you for detailed discussion. I checked your updated submission and the interpretations are now better. I want to follow-up on two points that are still not quite clear.
> >
> > **Regarding the fixed $|\gamma|$ ablation:** I don't think plotting the raw training loss in this ablation is the right approach. Smaller features admit higher values of training loss even if the training has converged perfectly, i.e. the $\epsilon$ parameter is exactly zero. Therefore, the correct quantity to plot is the excess loss $\epsilon$ itself, otherwise the theoretical lower-bound on the loss given fixed-sized features is a confounder.
> >
> > **Regarding the new Figure 6:** Here I am still a bit skeptical. If one pair of classes from 10 classes is tied together, that means that $20\%$ of all classes admit degenerate behavior. Thus, I am confused how can the predictor correctly distinguish and classify the samples from these two classes.
> >
> > The other points are in general resolved. Perhaps regarding the related work, while the current state is certainly an improvement over the past version, I would recommend the authors to double-check whether they didn't miss some other theory works on NC, since the literature is quite vast.

---

> > > ### Author Response · Authors · 2026-03-16
> > > **Thanks for your carefuly review and futher response to concerns**
> > >
> > > **Fixed $\|\gamma\|$ ablation — optimal loss gap $\epsilon$.**
> > >
> > > Thank you for this important observation. We agree that plotting the raw training loss is not the right quantity in the fixed-$|\gamma|$ ablation, since smaller feature norms can induce a strictly larger optimal loss even at convergence. In that case, the raw loss conflates optimization error with the theoretical lower bound associated with the constrained feature scale.
> > >
> > > Following your suggestion, we have revised Figure 4 to plot **optimal loss gap $\epsilon$**, defined as $L_{\mathrm{actual}} - m(\alpha\beta)$, where $\alpha$ is the average feature norm (which linearly correlates with the $|\gamma|$ value being controlled), $\beta$ is the recorded weight Frobenius norm at the end of the experiment, and $m(\alpha\beta)$ is the optimal loss value according to the formula in Theorem 2.1 using the computed $\alpha$ and $\beta$.
> > >
> > > Under this revised metric, the same qualitative trend persists: the loss gap $\epsilon$ decreases as $|\gamma|$ increases. This removes the confounding effect you pointed out and shows that the stronger NC observed at smaller $|\gamma|$ cannot be explained simply as a byproduct of faster convergence or better data fitting.
> > >
> > > For illustration, below is one example from the updated plot for the 4-layer conic-data experiment:
> > >
> > > | $\|\gamma_j\|=\frac{\|\gamma\|}{\sqrt{d}}$ | $\alpha\beta$ | $m(\alpha\beta)$ | Actual Loss | $\epsilon$ |
> > > |---|---|---|---|---|
> > > | 0.02 | 1.72 | 0.264 | 0.304 | 0.039 |
> > > | 0.05 | 3.00 | 0.053 | 0.086 | 0.032 |
> > > | 0.10 | 4.04 | 0.014 | 0.033 | 0.019 |
> > > | 0.20 | 5.14 | 0.003 | 0.013 | 0.009 |
> > > | 0.50 | 6.94 | 0.000 | 0.004 | 0.004 |
> > > | 1.00 | 5.88 | 0.000 | 0.002 | 0.002 |
> > >
> > >
> > > ---
> > >
> > > **Figure 6 — near-tied classes and classification behavior**
> > >
> > > Thank you for the follow-up question. We appreciate the opportunity to clarify this point further.
> > >
> > > We would like to clarify that, in these cases (CIFAR 100 experiments and VGG 19 without BN at higher WD), the resulting model **did not** classify almost all classes/samples perfectly and the losses **have not converged to be near-optimal**. In particular:
> > >
> > > 1. For the CIFAR 100 dataset, the accuracy for VGG11 and VGG19 with BN are < 75%, which is true even for official implementations. Considering that there are 100 classes, it's perfectly normal for the model to confuse between a few pair of classes and still achieve the accuracy at this level.
> > >
> > > 2. The typically used WD for the VGG networks is around 5e-4. Without BN, training with higher WD becomes much less stable. At a WD of 1e-2, the models without BN already fail to converge to near-perfect classification accuracy.
> > >
> > > We have added a note on this in the Figure 6 caption.
> > >
> > > As noted in the abstract and experiment section, the real-world experiments serve as complementary qualitative verification of the insights from our theoretical results, and we intentionally included settings beyond the strict assumptions of our theorem. The synthetic experiments in Figure 2, where near-optimal loss is achieved in nearly all cases, more directly illustrate the regime where our theoretical guarantees apply and are thus included in the main text.
> > >
> > > ---
> > >
> > > **Related work.**
> > >
> > > Thank you for the suggestion. We have conducted a more thorough review of the recent NC theory literature and added the following references as follows. If you have any further suggestions, we would gladly read them carefully and add them to our paper.
> > >
> > > - **Súkeník et al. (NeurIPS 2025), Wu & Mondelli (ICML 2025), Hong & Ling (2024), Wang et al. (2024), Guo et al. (TMLR 2025), Yan et al. (NeurIPS 2024), Wu & Papyan (NeurIPS 2024)**

---

> > > > ### Comment · Reviewer_oHQt · 2026-03-18
> > > >
> > > > Thank you for further clarifications. These new ablations and the explanation clarify my remaining concerns.

---

### Review · Reviewer_MfQc · 2026-02-23

**Summary Of Contributions:**

This paper investigates how weight decay (WD) and batch normalization (BN) influence the emergence of neural collapse (NC). Theoretically, the authors demonstrate that near the optimal regularized loss, core NC properties (zero within-class variation and maximal class separation) are satisfied asymptotically as the loss approaches its optimum or the weight decay parameter increases.

Empirically, the authors validate their theory by training deep models, with and without BN, across varying WD values. Their experiments show how the value of the training loss, the presence of BN, and the magnitude of WD impact NC metrics for within-class variation and class separation.

*Strengths*

The paper is well-structured, easy to follow, and generally well-written.

The analysis of the near-optimal regime and the derived bounds on the NC metrics are insightful and provide a good justification for how larger WD can lead to stronger NC convergence, which is also  empirically observed.

*Weaknesses*

In the theorems, the contribution of BN is not well articulated. A high-level proof sketch explaining how BN come into play makes this clearer. It would also be helpful to highlight the implications of removing BN in the theory.

The experiments do not fully verify all aspects of the theory. Apart from saying WD and BN are helpful for the emergence of NC, the theory provides a rate for how the near-optimality condition and WD affect the distance to NC. However, the experiments do not explicitly measure this. For instance, rather than simply plotting $intra_c$, plotting $1 - intra_c$ against the value of WD would help see if the theoretical decay rates wrt to WD are tight (also for $inter_{c,c'}$). Similarly, the relationship between the training loss value and the NC metrics in Figure 3 is currently only discussed as a *visual* correlation. For example, plotting $1 -intra_c$ versus the training loss value might provide a more rigorous check of this relationship.

**Audience:**

Yes

**Audience Explanation:**

The paper studies neural collapse, a recently discovered geometric property of last-layer representations in deep, over-parameterized networks. By using the abstract layer-peeled model, the authors provide a theoretical justification for the factors affecting this phenomenon, making the results relevant to TMLR's audience, particularly researchers focused on optimization and representation learning.

**Claims And Evidence:**

Yes

**Claims Explanation:**

The main theorems use standard assumptions and are supported by formal proofs. Empirically, the authors test models of varying architectures on synthetic and benchmark datasets. By manipulating BN and WD, the experiments back the central claims, although they do not exhaustively capture every theoretical detail.

**Requested Changes:**

1) The contributions section mentions a ``worst-class measure of NC''. Does the theory specifically guarantee a worst-case bound, or does it primarily state that the result holds asymptotically for the majority of the classes?

2) Can you explain why weight decay is not applied to the features in the theoretical setup and whether or not weight decay was also omitted for the features in the experiments?

3) I am slightly concerned if it is fair to compare the BN and no-BN settings while training both with the same learning rate and for the same number of steps. Since the theory focuses on the level of loss optimality and WD rates rather than convergence time, the models should ideally be trained to the same loss level before comparing their NC properties. In both Figures 3 and 7, in the majority of the cases, the models without BN have a larger loss value at the end of training compared to their BN counterpart.

*Minor*

4) Please define the layer-peeled model with BN rigorously early in the text for readers who might be unfamiliar with the abstraction.

5) Figure 3 needs formatting adjustments. It has overlapping y-axis values.

6) Section 3.4 seems to have a type: The first line claims that a higher feature norm leads to a stronger bound, but the end of the paragraph states that a lower norm ($\gamma$) induces stronger NC. There are also other minor typos in the appendix. For example, page 27 $CE_c(...)_{c(?)}$, page 28, softmax should be log(softmax(...)).

---

> ### Author Response · Authors · 2026-03-10
> **Response for MfQc review**
>
> Thank you for your detailed and constructive review. We have addressed each of your concerns below.
>
> **BN contribution not well articulated in theorems.**
>
> We have added a high-level proof sketch after Theorem 2.2. The key idea proceeds in three steps:
>
> 1. **BN enables norm control.** BN normalizes features to zero mean and unit variance along each dimension. The regularized scale parameter $\gamma$ then directly controls the post-normalization feature norms: specifically, $\|\mathbf{h}\|^2 \leq \|\gamma\|^2$ for batch-normalized features. Weight decay on $\gamma$ (with parameter $\lambda$) ensures $\|\gamma\|^2 \leq O(1/\lambda)$ at near-optimal loss, since otherwise the regularization penalty would dominate.
>
> 2. **Small norms imply strong local convexity.** The cross-entropy loss $-\log(\text{softmax}(\mathbf{z}))$ evaluated at logits $\mathbf{z} = \mathbf{W}\mathbf{h}$ has a local strong convexity constant that *increases* as the logit magnitudes decrease. When BN + WD keep $\|\mathbf{W}\|$ and $\|\mathbf{h}\|$ small, the loss landscape near the optimum becomes more strongly curved, meaning any near-optimal solution is tightly constrained.
>
> 3. **Strong convexity bounds NC deviations.** Using Jensen's inequality for strongly convex functions (Lemma 2.1), a small gap $\epsilon$ from the optimal loss forces the features to concentrate: intra-class features must be nearly collinear (NC1), and class means must approach the simplex ETF configuration (NC2). The rate $\rho = (Ce/\lambda)^{O(C)}$ captures how the WD parameter $\lambda$ controls the tightness of these bounds through the norm constraints.
>
> Without BN (or an equivalent feature norm regularizer), the feature norms are unconstrained even under WD, so step 1 breaks down and the bounds become vacuous.
>
> **Experiments don't verify theoretical rates.**
>
> Thank you for this helpful suggestion. We agree that a more direct empirical comparison with the theoretical bounds can improve the presentation.
>
>
> 1. **Verifying the perturbation rate:**  In the revised manuscript, we added plots (appendix Figure 10, NC metrics vs training loss scatter plot across models; appendix Figure 11, NC deviation vs training loss on a log-log scale for individual training runs) that explicitly measure the deviation from NC compared to the loss gap. In particular, we now report the quantities 1-intra and the deviation of inter-class cosine similarities from their ETF value, and plot them against the training loss on a log-log scale (Figure 11) and direct scatter plot among experiments (Figure 10), which we use as a proxy for the optimality gap $\epsilon$. As shown in the figures, the deviation follows a power-law relationship with $\epsilon$, consistent with the scaling predicted by Theorem 2.2.
>
>
> 2. **Dependence on weight decay:** The dependence on $\lambda$ enters the bound through the multiplicative factor $\rho$. Empirically, we also examine the dependence on $\lambda$, and we observe that a stronger WD is associated with a smaller NC deviation (Figure 2, NC vs WD plot), particularly when combined with norm control through BN. This qualitative trend is consistent with the theoretical dependence through $\rho$.
>
>
> However, since $\rho$ arises from conservative worst-case asymptotic estimates in the analysis, we do not expect the complicated precise $\lambda$-dependence in the bound to appear as a tight empirical rate. For this reason, we interpret the WD plots primarily as a qualitative consistency check rather than a tight rate verification.
>
> 3. **Dependence on feature norm:** As shown in Figure 4 (fixed-$|\gamma|$ experiments), smaller feature norms lead to stronger NC at comparable loss. This observation is consistent with the norm-control interpretation of the theoretical bounds, where tighter norm constraints reduce the admissible deviation from the NC geometry.
>
>
> 4. **Complementary roles of theory and experiments:** More broadly, we view the theoretical and experimental results as complementary. The theory establishes a perturbation result showing that under norm control (as induced by WD and regularized BN), near-optimal solutions must lie close to the NC geometry. However, the theorem itself does not analyze counterfactual settings such as the absence of BN.
>
> The experiments, therefore, play a complementary role: they investigate how practical training mechanisms such as BN and WD affect the empirical emergence and strength of NC. In particular, we observe that NC metrics become substantially stronger and more monotonic when BN-based norm control is present. This complements the theoretical result by illustrating *qualitatively how such norm-control mechanisms manifest in realistic training dynamics rather than empirical justification of the quantitative values.

---

> ### Author Response · Authors · 2026-03-10
> **Response for MfQc review (part 2)**
>
> **"Worst-class measure" --- worst-case or asymptotic majority?**
>
> The theory guarantees bounds for a **$(1-\delta)$ fraction of classes** (for NC1/NC3) and a **$(1-\delta)$ fraction of class pairs** (for NC2), where $\delta$ is a free parameter that appears in the bound denominators. Notably, if we take $\delta=1-\frac{1}{c}$ in NC1/3 and $\delta=1-\frac{2}{C(C-1)}$ (i.e., 1-1/(number of class pairs)), we get the worse case single class & class pair bounds
> We have revised the contributions section to replace "worst-class measure" with "worst-class and high-probability per-class measure" and explicitly state the $(1-\delta)$ qualification. Our experiments report both the average and worst-class metrics, and we observe that the worst class typically follows the same qualitative trends as the average.
>
> **Why is WD not applied to features in theory/experiments?**
>
> In the layer-peeled model, features $\mathbf{h}$ are free optimization variables representing the output of all hidden layers. In a real network, WD is applied to the *weights* of each layer, not directly to the activations/features. The features are only indirectly regularized through the weight regularization of earlier layers.
>
> In our theoretical setup, we apply WD to: (1) the last-layer weight matrix $\mathbf{W}$, and (2) the BN scale parameters $\gamma$. We do not apply WD directly to $\mathbf{h}$ because $\mathbf{h}$ is not a network parameter; it is an activation. However, regularizing $\gamma$ after BN normalization *effectively* constrains $\|\mathbf{h}\|$, as we now make explicit in Section 2.1. This mirrors real training: the feature norms are controlled through the BN parameters, not through direct feature regularization.
>
> In our experiments, WD is applied to all network parameters (weights and BN parameters) via the optimizer, consistent with standard PyTorch practice.
>
> **Fair comparison --- BN vs no-BN at same LR/steps.**
>
> This is a valid concern. We address this in two ways:
>
> 1. **Per-epoch matched-loss analysis:** Using per-epoch checkpoints (every 5 epochs), we compare NC metrics between BN and no-BN models at the *same training loss level* (see appendix Figure 10). At any given loss level, BN models consistently exhibit stronger NC than no-BN models. This confirms that BN's NC advantage is not merely a consequence of achieving lower loss.
>
> 2. **Theoretical justification:** Our bounds (Theorem 2.2) indicate that at a given loss level $\epsilon$, BN's norm control (via regularized $\gamma$) yields bounded NC deviations — which is exactly what the matched-loss comparison confirms. Without BN, feature norms are unconstrained, so the bounds do not apply.
>
> **Minor:**
>
> 1. We have added a concise definition of the layer-peeled model (also known as the unconstrained features model) in Section 1.1, clarifying that hidden-layer parameters are treated as fixed while optimization is performed only over the last-layer weights $\mathbf{W}$ and features $\mathbf{h}$ as free variables.
>
> 2. We have regenerated Figure 3 with dual y-axes (cosine similarity on the left and loss on a log-scale right axis) and improved subplot spacing to eliminate the overlapping labels.
>
> 3. We have corrected the softmax notation so that `softmax(...)` now correctly appears as `-log(softmax(...))` throughout the proofs.

---

### Author Response · Authors · 2026-03-10
**General Response to Reviewers**

We thank the reviewers for their careful reading and constructive feedback. The reviews have substantially improved the clarity and rigor of the manuscript. In particular, we appreciate that the reviewers recognized the value of studying Neural Collapse through the lens of perturbation analysis and the potential contributions of connecting BN and WD to the NC geometry.

Several themes emerged across the reviews, and we address them here before responding to each reviewer individually.

**Clarifying the scope of the theoretical contribution (raised by Reviewers oHQt and eHk8).**

A common concern was that the theory could be read as establishing a causal mechanism by which WD or BN produce NC during training, which our theory does not provide sufficient evidence for. We have revised the manuscript to make the scope more precise:

- *Theoretical contribution.* Our theorems are perturbation bounds: for any solution within an $\epsilon$-optimality gap of the regularized CE loss, the deviation of the NC metrics is bounded as a function of $\epsilon$ and the norm-control parameters. The key role of WD and regularized BN in the theory is to provide feature norm control, which tightens these bounds at a fixed loss level.

- *Empirical contribution.* Our experiments are complementary empirical findings rather than direct verifications of the theory. They show how BN and WD affect the emergence, monotonicity, and strength of NC in practical training, including settings such as no-BN that are not directly covered by the theory.

To reflect this distinction, we revised the abstract, introduction, and theorem discussion to consistently frame the theory as a perturbation analysis and the experiments as complementary empirical evidence.

**BN as feature norm regularization (raised by Reviewers oHQt and MfQc).**

We have made explicit in Section 2.1 that when BN scale parameters are learnable and regularized, the combined effect is feature norm control — BN normalizes to unit variance, and regularized $\gamma$ controls the post-normalization scale. This is the operative mechanism in our bounds. The theory addresses the setting with this norm control; the no-BN experiments provide complementary empirical evidence about what happens without it.

**Theory-experiment connection (raised by all reviewers).**

The theoretical bounds include architecture-specific constants that are not tight, preventing exact quantitative rate verification. We have added the following experiments that examine the qualitative trends:

1. **NC vs. WD (Figure 2):** BN models show monotonically improving NC with increasing $\lambda$. No-BN models show weaker and less consistent trends.

2. **NC vs. training loss (appendix Figure 10):** Per-epoch scatter plots over different experiments show a strong monotonic NC-loss relationship for BN models. No-BN models show weaker correlation.

3. **NC deviation vs. loss (appendix Figure 11):** Worst-class NC deviations plotted against training loss on a log-log scale for individual training runs, showing the power-law relationship between optimality gap and NC proximity.

4. **Convergence ablation (Figure 4 with added final loss comparison):**
For the fixed-$|\gamma|$ experiments (Section 3.4), we report the final training loss together with the NC metrics. A potential concern is that smaller feature norms might simply allow faster optimization or lead to lower final loss, which could artificially strengthen NC. The added loss comparison shows the opposite trend: larger feature norms achieve lower training loss but exhibit weaker NC. This rules out the possibility that the stronger NC observed at smaller $|\gamma|$ is merely a byproduct of faster convergence or better data fitting.

5. **NC3 self-duality:**
Following the reviewer’s suggestion, we have added NC3 self-duality plots (Figure 5) to the main text, which track the alignment between classifier weights and class means during training.

**Related work updated (raised by Reviewers oHQt and eHk8).**

We have updated the Related Work section with recent references: Súkeník et al. (NeurIPS 2024), Dang et al. (ICML 2024), Hong and Ling (JMLR 2024), Jacot et al. (ICLR 2025), and Munn et al. (NeurIPS 2024).

---

### Decision · Action_Editor_C6am · 2026-04-07

**Recommendation:** Accept as is

**Additional Comments:**

This paper studies the emergence of neural collapse connecting it to batch normalization and weight decay. On the theoretical side, the authors show how much a near-optimal solution of the unconstrained feature model deviates from collapse. This deviation is shown to depend on the weight decay. On the experimental side, the authors demonstrate the impact of batch normalization in establishing neural collapse and discuss the interaction between batch normalization and weight decay.

All reviewers found the results interesting enough for TMLR's readership (albeit with varying levels of enthusiasm), but raised a number of criticisms to the way in which the results were framed and interpreted, as well as on the weak links between theory and experiments. The authors have provided a comprehensive rebuttal and revised the paper accordingly. In particular, they have clarified the scope of the theoretical contribution (revising abstract, introduction and theorem discussion) and added more experimental results and ablations. More generally, all outstanding concerns have been resolved and, for this reason, I am happy to recommend acceptance.

**Audience:**

Yes

**Audience Explanation:**

Neural Collapse is a topic of interest to the TMLR readership and this paper makes interesting contributions both on the theoretical and on the experimental fronts.

**Claims And Evidence:**

Yes

**Claims Explanation:**

While the reviewers expressed concerns about the claims made in the original version of the paper, such concerns have been addressed in a satisfactory way during the rebuttal and the claims made in the revision are well supported.